# Measuring Mechanistic Interpretability at Scale Without Humans

**Roland S. Zimmermannl**[*]
MPI-IS, Tübingen AI Center

**David Klindt**
Stanford University

**Wieland Brendel**
MPI-IS, Tübingen AI Center

## Abstract

In today's era, whatever we can measure at scale, we can optimize. So far, measuring the interpretability of units in deep neural networks (DNNs) for computer vision still requires direct human evaluation and is not scalable. As a result, the inner workings of DNNs remain a mystery despite the remarkable progress we have seen in their applications. In this work, we introduce the first scalable method to measure the per-unit interpretability in vision DNNs. This method does not require any human evaluations, yet its prediction correlates well with existing human interpretability measurements. We validate its predictive power through an interventional human psychophysics study. We demonstrate the usefulness of this measure by performing previously infeasible experiments: (1) A large-scale interpretability analysis across more than 70 million units from 835 computer vision models, and (2) an extensive analysis of how units transform during training. We find an anticorrelation between a model's downstream classification performance and per-unit interpretability, which is also observable during model training. Furthermore, we see that a layer's location and width influence its interpretability.

## 1 Introduction

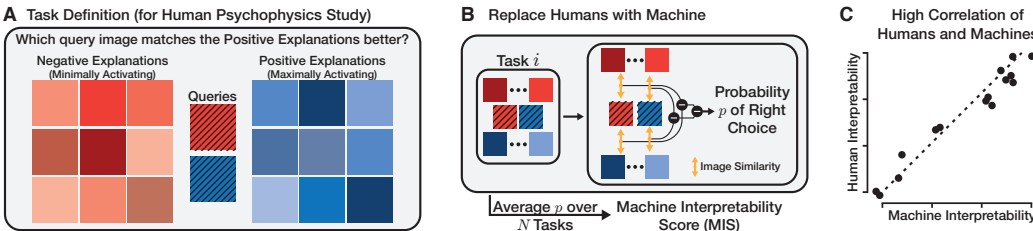

Fig. 1: **Definition of the Machine Interpretability Score. A.** We build on top of the established task definition proposed by Borowski et al. (2021) to quantify the per-unit interpretability via human psychophysics experiments. The task quantifies how well participants understand the sensitivity of a unit by asking them to match strongly activating query images to strongly activating *visual* explanations of the unit. See Fig. 9 for examples. **B.** Crucially, we remove the need for humans and fully automate the evaluation: We pass the explanations and query images through a feature encoder to compute pair-wise image similarities (DreamSim) before using a (hard-coded) binary classifier to solve the underlying task. Finally, the Machine Interpretability Score (MIS) is the average of the predicted probability of the correct choice over $N$ tasks. **C.** The MIS proves to be highly correlated with human interpretability ratings and allows fast evaluations of new hypotheses.

With the arrival of the first non-trivial neural networks, researchers got interested in understanding their inner workings (Krizhevsky et al., 2012; Mahendran & Vedaldi, 2015). For one, this can be motivated by scientific curiosity; for another, a better understanding might lead to building more reliable, efficient, or fairer models. While the performance of neural networks has seen a remarkable improvement over the last few years, our understanding of information processing has progressed more slowly. Nevertheless, understanding how complex models — e.g., language models (Bricken

---

[*]Project website: brendel-group.github.io/mis. Correspondence: research@rzimmermann.com

et al., 2023) or vision models (Olah et al., 2017; Zimmermann et al., 2023) — work is still an active and growing field of research, coined *mechanistic interpretability* (Olah, 2022). A common approach in this field is to divide a network into atomic units, hoping they are easier to comprehend. Here, atomic units might refer to individual neurons or channels of (convolutional) layers (Olah et al., 2017), or general vectors in feature space (Elhage et al., 2022; Klindt et al., 2023). Besides this approach, mechanistic interpretability also includes the detection of neural circuits (Cammarata et al., 2020; Elhage et al., 2022) or analysis of global network properties (Nanda et al., 2023).

The goal of understanding the inner workings of a neural network is inherently human-centric: Irrespective of what tools have been used, in the end, humans should have a better comprehension of the network. However, human evaluations are time-consuming and costly due to their reliance on human labor (Zimmermann et al., 2023). This results in slower research progress, as validating novel hypotheses takes longer. Removing the need for human labor by automating the evaluation can open up multiple high-impact research directions: One benefit is that it enables the creation of more interpretable networks by explicitly optimizing for interpretability — after all, what we can measure at scale, we can optimize. Moreover, it allows more efficient research on explanation methods and might increase the overall understanding of neural networks. While efforts to build such measures for language models exist (Bills et al., 2023), there is no common approach yet for vision models.

The present work is the first to introduce a fully automated interpretability measure (Fig. 1A & C): the Machine Interpretability Score (MIS). By leveraging the latest advances in image similarity functions aligned with human perception, we obtain a measure that is strongly predictive of human-perceived interpretability (Fig. 1C). We verify our measure through both correlational and interventional experiments. By removing the need for human labor, we can scale up existing evaluations by multiple orders of magnitude. Finally, we demonstrate potential workflows and use cases of our MIS.

## 2 RELATED WORK

**Mechanistic Interpretability**  While the overall field of explainable AI tries to increase our understanding of neural networks, multiple subbranches with different foci exist (Gilpin et al., 2018). One of these branches, *mechanistic interpretability*, aims to improve our understanding of neural networks by understanding their building blocks (Olah, 2022). An even more fine-grained branch aims to interpret individual units of vision models (Bau et al., 2017; Zhou et al., 2018; Bau et al., 2020; Morcos et al., 2018; Olah et al., 2017). We focus exclusively on this branch of research. This line of research for artificial neural networks was, arguably, inspired by similar efforts in neuroscience for biological neural networks (Hubel & Wiesel, 1962; Barlow, 1972; Quiroga et al., 2005).

Different studies set out to understand the behavior and sensitivity of individual units of vision networks – here, a unit can, e.g., be (the spatial average of) a channel in a convolutional neural network (CNN) or a neuron in a multilayer perceptron (MLP). The level of understanding obtained for a unit is commonly called the *per-unit interpretability*; by averaging over a representative subset of units in the network, one obtains the *per-model interpretability* (Borowski et al., 2021; Zimmermann et al., 2023). With the recent progress in vision-language modeling, a few approaches started using textual descriptions of a unit's behavior (Hernandez et al., 2022; Kalibhat et al., 2023). However, the majority still uses visual explanations which are either synthesized by performing activation maximization through, e.g., gradient ascent (Olah et al., 2017; Erhan et al., 2009; Mahendran & Vedaldi, 2015; Nguyen et al., 2014; Mordvintsev et al., 2015; Yosinski et al., 2015; Nguyen et al., 2017), or strongly activating dataset examples (Olah et al., 2017; Borowski et al., 2021).

With the onset of large language models (LLMs) and the increasing interest in them, there is also now an increasing interest in mechanistic interpretability of them (e.g., Elhage et al., 2021; Olsson et al., 2022; Bricken et al., 2023).

**Quantifying Interpretability**  Rigorous evaluations, including falsifiable hypothesis testing, are critical for research on interpretability methods (Leavitt & Morcos, 2020). This also encompasses the need for human-centric evaluations (Borowski et al., 2021; Kim et al., 2022). Nevertheless, such human-centric evaluations of interpretability methods are only available in some sub-fields. Specifically for the type of interpretability we are concerned about in this work, i.e., the per-unit interpretability of vision models, two methods for quantifying the helpfulness of explanations to humans were introduced before: Borowski et al. (2021) presented a two-alternative-forced-choice

(2-AFC) psychophysics task that requires participants to determine which of two images elicits higher activation of the unit in question, given visual explanations (i.e., images that strongly activate or deactivate the unit, see Fig. 1A) of the unit's behavior. Zimmermann et al. (2021) extended this paradigm to quantify how well participants can predict the influence of interventions in the form of occlusions in images. While these studies used their paradigms to evaluate the usefulness of different interpretability methods, Zimmermann et al. (2023) leveraged them to compare the interpretability of multiple models. Due to the reliance on human experiments, they could only probe the interpretability of 767 units from nine models. We now automatize this evaluation to scale it up by multiple orders of magnitude to more than 70 million units across 835 models.

**Automating Interpretability Research**    To increase the efficiency of interpretability research and scale it to large modern-day networks, the concept of automated interpretability was proposed, first in the domain of natural language processing (Bills et al., 2023). This approach uses an LLM to generate textual descriptions of the behavior of units in another LLM. Follow-up work by Huang et al. (2023), however, pointed out potential problems regarding the correctness of the explanations. To benchmark future fully automated interpretability tools, acting as independent agents, Schwettmann et al. (2023) introduced a synthetic benchmark suite inspired by the behavior of neural networks. In computer vision, there are also efforts to automate interpretability research (Hernandez et al., 2022; Zimmermann et al., 2023). Hernandez et al. (2022) and Oikarinen & Weng (2022) map visual to textual explanations of a unit's behavior using automated tools, hoping to increase the efficiency of evaluations. Zimmermann et al. (2023) introduced the ImageNet Mechanistic Interpretability (IMI) dataset, containing per-unit interpretability annotations from humans for 767 units, meant to foster research on automating interpretability evaluations.

## 3    METHOD

We now introduce our fully automated interpretability measure, Machine Interpretability Score (MIS), visualized in Fig. 1. Borowski et al. (2021) proposed a psychophysical experiment for quantifying the per-unit interpretability of vision models, i.e., how well humans can infer the sensitivity of a unit in a vision model from visual explanations. Here, a unit can be a channel in a CNN, commonly averaged over space, a neuron in an MLP, or arbitrary linear combinations of different units. The experiment uses a 2-AFC task design (see Fig. 1A) to measure how well humans understand a unit by probing how well they can predict which of two extremely activating (query) images yields a higher activation, after seeing visual explanations. Specifically, two sets of explanations are displayed: highly and weakly activating images, called positive and negative explanations, respectively. See Appx. A.1 for a more detailed task description. We build on top of this paradigm but replace human participants with machines, resulting in a fully automated interpretability metric that requires no humans.

**Definition of the Machine Interpretability Score**    Let $\mathcal{I}$ denote the space of valid input images for a model. For a specific explanation method and a unit in question, we denote the unit's positive and negative visual explanations as sets of images $\mathcal{E}^+ \subseteq \mathcal{I}$ and $\mathcal{E}^- \subseteq \mathcal{I}$, respectively. Further, let $\mathcal{Q}^+ \subseteq \mathcal{I}$ and $\mathcal{Q}^- \subseteq \mathcal{I}$ be the sets of query images with the most extreme (positive and negative) activations. The task by Borowski et al. (2021) can now be expressed as: Given explanations $\mathcal{E}^+$ and $\mathcal{E}^-$ and two queries $\mathbf{q}^+ \in \mathcal{Q}^+$ and $\mathbf{q}^- \in \mathcal{Q}^-$, which of the two queries matches $\mathcal{E}^+$ and which $\mathcal{E}^-$ more closely? An intuitive way to solve this binary task is to compare each query with every explanation and to match the query images to the sets of explanations based on the images' similarities.

To formalize this, we introduce a perceptual (image) similarity function $f : \mathcal{I} \times \mathcal{I} \to \mathbb{R}$ computing the scalar similarity of two images (Zhang et al., 2018), and an aggregation function $a : \mathbb{R}^K \to \mathbb{R}$ reducing a set of $K$ similarities to a single one. This allows us to define the function $s : \mathcal{I} \times \mathcal{I}^K \to \mathbb{R}$ that quantifies the similarity of a single query image to a set of explanations:

$$s(\mathbf{q}, \mathcal{E}) := a\left(\{\, f(\mathbf{q}, \mathbf{e}) \mid \mathbf{e} \in \mathcal{E}\,\}\right). \tag{1}$$

To decide whether a single query image is more likely to be the positive one, we can compute whether it is more similar to the positive than the negative explanations. We can compute this now for both the positive and the negative query images and get:

$$\Delta_+(\mathbf{q}^+, \mathcal{E}^+, \mathcal{E}^-) = s(\mathbf{q}^+, \mathcal{E}^+) - s(\mathbf{q}^+, \mathcal{E}^-), \tag{2}$$

$$\Delta_-(\mathbf{q}^-, \mathcal{E}^+, \mathcal{E}^-) = s(\mathbf{q}^-, \mathcal{E}^+) - s(\mathbf{q}^-, \mathcal{E}^-). \tag{3}$$

The classification problem will be solved correctly if the similarity of $\mathbf{q}^+$ to $\mathcal{E}^+$ relative to $\mathcal{E}^-$ is stronger than those of $\mathbf{q}^-$. This means we can define the probability of solving the binary classification problem correctly as

$$p(\mathbf{q}^+, \mathbf{q}^-, \mathcal{E}^+, \mathcal{E}^-) := \sigma\Big(\alpha \cdot \big(\Delta_+(\mathbf{q}^+, \mathcal{E}^+, \mathcal{E}^-) - \Delta_-(\mathbf{q}^-, \mathcal{E}^+, \mathcal{E}^-)\big)\Big) \tag{4}$$

where $\sigma$ denotes the sigmoid function and $\alpha$ is a free parameter to calibrate the classifier's confidence.

We define the *Machine Interpretability Score* (MIS) as the predicted probability of making the right choice, averaged over $N$ tasks for the same unit. Across these different tasks, the query images $\mathbf{q}^+, \mathbf{q}^-$ vary to cover a wider range of the unit's behavior. If the explanation method used is stochastic, it is advisable to also average over different explanations:

$$\text{MIS} = \frac{1}{N}\sum_i^N p(\mathbf{q}_i^+, \mathbf{q}_i^-, \mathcal{E}_i^+, \mathcal{E}_i^-). \tag{5}$$

The MIS is not a general property of a unit but depends on the method used to generate explanations. One might define a general score by computing the maximum MIS over multiple explanation methods.

**Choice of Hyperparameters.** We use the current state-of-the-art perceptual similarity, DreamSim (Fu et al., 2023), as $f$. See Appx. B for a sensitivity study on this choice. DreamSim models the perceptual similarity of two images as the cosine similarity of the images' representations from (multiple) computer vision backbones. These were first pre-trained with, e.g., CLIP-style training (Radford et al., 2021) and then fine-tuned to match human annotations for image similarities of pairs of images. We use the mean to aggregate the distances between a query image and multiple explanations to a single scalar, i.e., $a(x_1, \ldots, x_K) := 1/K \sum_i^K x_i$. To choose $\alpha$, we use the interpretability annotations of IMI Zimmermann et al. (2023): We optimize $\alpha$ over a randomly chosen subset of just 5% of the annotated units to approximately match the value range of human interpretability scores, resulting in $\alpha = 0.16$. Note that $\alpha$ is, in fact, the only free parameter of our metric, resulting in very low chances of overfitting the metric to the IMI dataset. We use the same strategy as Borowski et al. (2021); Zimmermann et al. (2021) and Zimmermann et al. (2023) for generating new tasks (see Appx. A.2). As they used up to 20 tasks per unit, we average over $N = 20$. See Appx. C for a sensitivity study.

## 4 RESULTS

This section is structured into two parts: First, we validate our Machine Interpretability Score (MIS) by showing that it is well correlated with existing interpretability annotations. Then, we demonstrate what type of experiments become feasible by having access to such an automated interpretability measure. Our experiments use the best-working — according to human judgements (Borowski et al., 2021) — visual explanation method, dataset examples, for computing the MIS. We demonstrate the applicability of our method to other interpretability methods (e.g., feature visualizations) in Appx. D. Note that different explanation methods might require different hyperparameters for computing the MIS. Both query images and explanations are chosen from the training set of ImageNet-2012 (Russakovsky et al., 2015). When investigating layers whose feature maps have spatial dimensions, we consider the spatial mean over a channel as one unit (e.g., Borowski et al., 2021). We ignore units with constant activations from our analysis as there is no behavior to understand (see Appx. E for details).

### 4.1 VALIDATING THE MACHINE INTERPRETABILITY SCORE

We validate our MIS measure by using the interpretability annotations in the IMI dataset (Zimmermann et al., 2023), which will be referred to as Human Interpretability Scores (HIS). The per-unit annotations are responses to the 2-AFC task described in Sec. 3, averaged over $\approx 30$ participants. IMI contains scores for a subset of units for nine models.[1]

---

[1] Zimmermann et al. (2023) investigate nine different models but test two of them in multiple settings, resulting in 14 distinct experimental conditions to compare.

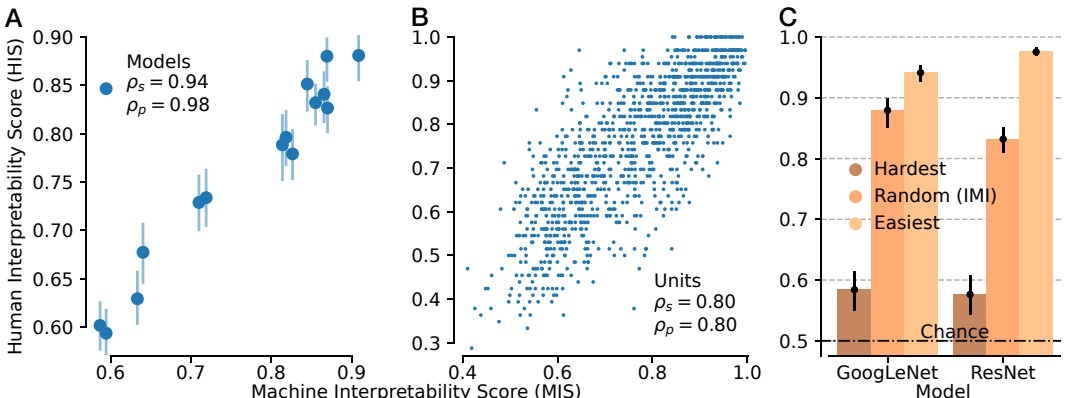

Fig. 2: **Validation of the MIS.** Our proposed Machine Interpretability Score (MIS) explains existing interpretability annotations (Human Interpretability Score, HIS) from IMI (Zimmermann et al., 2023) well. **(A) MIS Explains Interpretability Model Rankings.** The MIS reproduces the ranking of models presented in IMI while being fully automated and not requiring any human labor, as evident by the strong correlation between MIS and HIS. Similar results are found for the interpretability afforded by another explanation method in Appx. D. **(B) MIS Explains Per-unit Interpretability Annotations.** The MIS also explains individual per-unit interpretability annotations. We show the calculated MIS and the recorded HIS for every unit in IMI and find a high correlation matching the noise ceiling at $\rho = 0.80$ (see Appx. B). **(C) MIS Allows Detection of (Non-) Interpretable Units.** We use the MIS to perform a causal intervention and determine the least (*hardest*) and most (*easiest*) interpretable units in a GoogLeNet and ResNet-50. Using the psychophysics setup of Zimmermann et al. (2023), we measure their interpretability and compare them to randomly sampled units. Strikingly, the psychophysics results match the predicted properties: Units with the lowest MIS have significantly lower interpretability than random units, which have significantly lower interpretability than those with the highest MIS. Errorbars denote the 95 % confidence interval.

#### 4.1.1 MIS EXPLAINS EXISTING DATA

First, we reproduce the main result of Zimmermann et al. (2023): A comparison of nine models in terms of their the per-unit interpretability. We plot the HIS and MIS values (averaged over all units in a model) in Fig. 2A and find very strong correlations (Pearson's $r = 0.98$ and Spearman's $r = 0.94$). Reproducing the model ranking is strong evidence for the validity of the metric, as no information about these rankings was explicitly used to create our new measure.

Next, we can zoom in and look at individual units instead of per-model averages. Fig. 2B shows MIS and HIS for all units of IMI. The left figure clearly shows a strong correlation (Pearson's and Spearman's $r = 0.80$). The interpretability scores in IMI are a (potentially noisy) estimate over a finite number of annotators. We estimate the ceiling performance due to noise (sampling 30 trials from a Bernoulli distribution) to equal a Pearson's $r = 0.82$ (see Appx. B for details). The right figure shows an alternative visualization, which bins the units according to their MIS and averages the HIS to reduce this noise — highlighting that the two scores correlate strongly. We can conclude that the MIS explains existing interpretability annotations well - both on a per-unit and per-model level.

#### 4.1.2 MIS MAKES NOVEL PREDICTIONS

While the previous results show a strong relation between MIS and human-perceived interpretability, they are of a descriptive (correlational) nature. To further test the match between MIS and HIS, we now turn to a causal (interventional) experiment: Instead of predicting the interpretability of units *after* a psychophysics evaluation produced their human scores, we now compute the MIS *before* conducting the psychophysics evaluation. We perform our experiment for two models: GoogLeNet and a ResNet-50. For each model, IMI contains interpretability scores for 96 randomly chosen units. We look at all the units not tested so far and find the 42 units yielding the highest (Easiest, average of 0.99 for both models) and lowest (Hardest, average of 0.63 and 0.59, respectively) MIS, respectively. Then, we use the same setup as Zimmermann et al. (2023) and perform a psychophysical evaluation

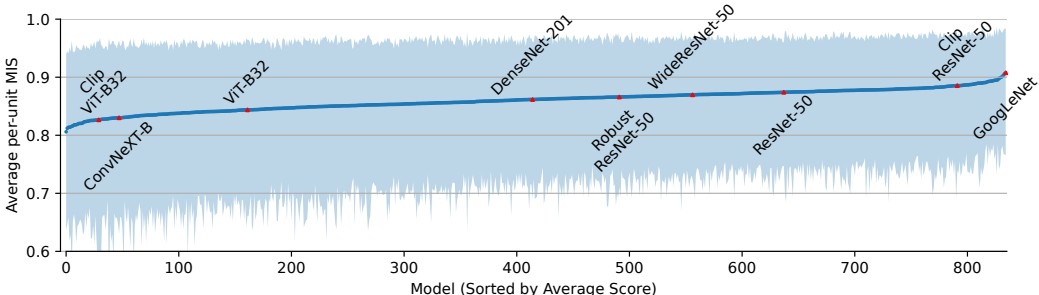

Fig. 3: **Comparison of the Average Per-unit MIS for Models.** We substantially extend the analysis of Zimmermann et al. (2023) from a noisy average over a few units for a few models to all units of 835 models. The models are compared regarding their average per-unit interpretability (as judged by MIS); the shaded area depicts the 5th to 95th percentile over units. We see that all models fall into an intermediate performance regime, with stronger changes in interpretability at the tails of the model ranking. Models probed by Zimmermann et al. (2023) are highlighted in red.

on Amazon Mechanical Turk with 236 participants. We compare the (newly) recorded HIS for the three groups of units in Fig. 2C. The results are very clear again: As predicted by the MIS, the HIS is highest for the easiest and lowest for the hardest units. Further, the HIS is close to the pre-hoc determined MIS given above. This demonstrates the strong predictive power of the MIS and its ability to be used for formulating novel hypotheses.

## 4.2 ANALYZING & COMPARING HUNDREDS OF MODELS

After confirming the validity of the MIS, we now change gears and show use cases for it, i.e., experiments and analyses that were truly infeasible before due to the high cost, both time and money, of human evaluations.

### 4.2.1 COMPARISON OF MODELS

Zimmermann et al. (2023) investigated whether model or training design choices influence the interpretability of vision models. Although they invested a considerable amount of money in this investigation ($\geq 12\,000$ USD), they could only compare nine models via a subset of units. We now scale up this line of work by two orders of magnitude and investigate all units of 835 models, almost all of which come from the well-established computer vision library timm (Wightman, 2019). These models differ in architecture and training datasets but were all at least fine-tuned on ImageNet. See Appx. F for a list of models. Putting this scale into perspective, achieving the same scale by scaling up previous human psychophysics experiments would amount to absurd costs ($\geq 1$ B USD). Following previous work, we ignore the first and last layers of each model (Zimmermann et al., 2023).

When sorting the models according to their average MIS (Fig. 3) they span a value range of $\approx 0.80 - 0.91$. The strongest differences across models are present at the tails of the ranking. Note that GoogLeNet is ranked as the most interpretable model, resonating with the community's interest in GoogLeNet as it is widely claimed to be more interpretable. The shaded area denotes the 5th to 95th percentile of the distribution across units. This reveals a strong difference in the variability of units for different models; further, as the upper end of the MIS is similar across models ($\approx 95\,\%$), most of the change in the average score seems to stem from a change in the lower end, with decreasing width of the per-unit distribution for higher model rank.

To investigate the difference in how the MIS of units is distributed between different models, we select 15 exemplary models and visualize their per-unit MIS distribution in Fig. 4B. Those models were chosen according to the distance between 5th and 95th percentile (five with highest, average, and lowest distance). While models with low and medium variability have unimodal left-skewed distributions, the ones with high variability have a rather bimodal distribution. Note that the distribution's second, stronger mode has a similar mean and shape to the overall distribution for models with low variability. The first mode is placed at a value range slightly above 0.5, corresponding to the task's chance level, indicating mostly uninterpretable units. This suggests that a subset of uninterpretable

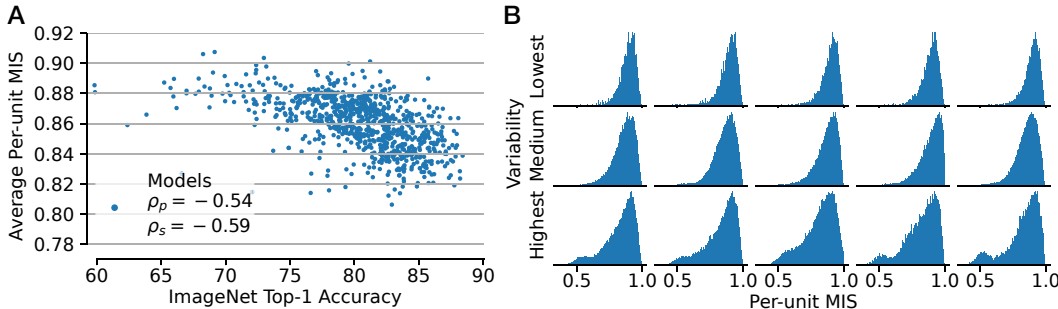

Fig. 4: **(A) Relation Between ImageNet Accuracy and MIS.** The average per-unit MIS of a model is anticorrelated with the model's ImageNet classification accuracy. **(B) Distribution of per-unit MIS.** Distribution of the per-unit MIS for 15 models, chosen based on the size of the error bar in Fig. 3: lowest (top), medium (middle), and highest variability (bottom row). While most models have an unimodal distribution, those with high variability have a second mode with lower MIS.

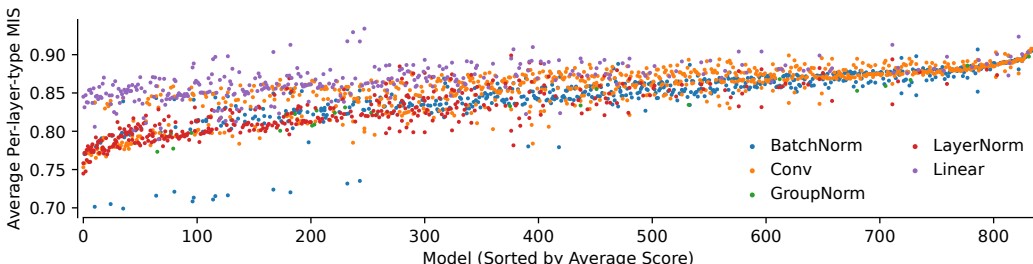

Fig. 5: **Comparison of the Average Per-unit MIS for Different Layer Types and Models.** We show the average interpretability of units from the most common layer types in vision models (BatchNorm, Conv, GroupNorm, LayerNorm, Linear). We follow Zimmermann et al. (2023) and restrict our analysis of Vision Transformers to the linear layers in each attention head. While not every layer type is used by every model, we still see some separation between types (see Fig. 16 for significance results): Linear and convolutional layers mostly outperform normalization layers. Models are sorted by average per-unit interpretability, as in Fig. 3.

units (see Fig. 22 for examples) can explain most of the models' differences in average MIS. We analyze this further in Fig. 19, where we compare the models in terms of their worst units. We see a similar shape as in Fig. 3, but with a larger value range used, resulting in stronger model differences.

Previous work analyzed a potential correlation between interpretability and downstream classification performance. However, in a limited evaluation, it was found that better classifiers are not necessarily more interpretable (Zimmermann et al., 2023). A re-evaluation of this question is performed in Fig. 4A and paints an even darker picture: Here, better performing ImageNet classifiers are less interpretable (Pearson's $r = -0.5$ and Spearman's $r = -0.55$).

Among training procedures and architecture, the analyzed models also differ in the required resolution of their input. While previous work focused only on models with a single resolution (Zimmermann et al., 2023), we can now see whether the resolution influences interpretability. However, Fig. 17 suggests that there is no influence.

### 4.2.2 COMPARISON OF LAYERS

Next, we can zoom into the results of Fig. 3 and investigate whether there are differences between different layers. First, we are interested in testing whether the layer type is important, e.g., are convolutional more interpretable than normalization or linear layers? In Fig. 5, we sort the models by their average MIS over all layer types but show individual points for each of the five most common types (Conv, Linear, BatchNorm, LayerNorm, and GroupNorm). The number of points per model may vary, as not all models contain layers of all types. The figure shows a benefit of Conv over

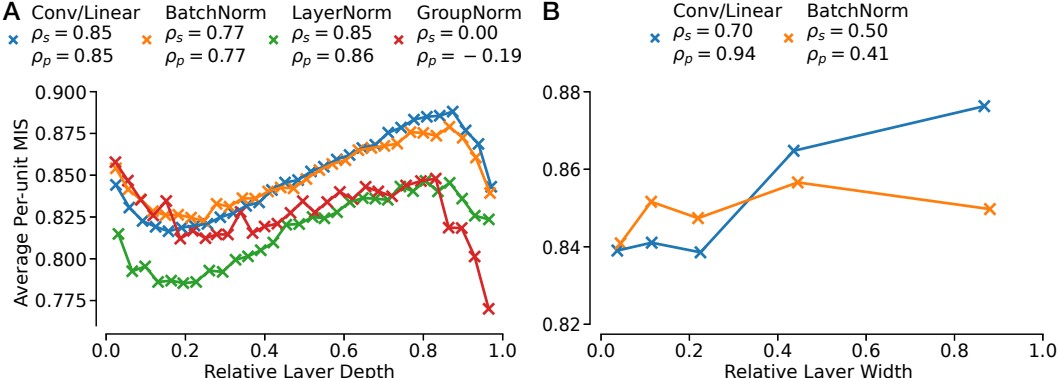

Fig. 6: **(A) Deeper Layers are More Interpretable.** Average MIS per layer as a function of the relative depth of the layer within the network, grouped by layer types. For type, the values are grouped into 30 bins of equal count based on the relative depth; values shown correspond to the bin average. **(B) Wider Layers are More Interpretable.** Average MIS per layer as a function of the relative width of the layer compared to all layers of the same type in the network, grouped by layer types. For each type, the values are grouped into 5 bins, as above.

BatchNorm layers, which themselves are better than LayerNorm layers. Linear layers, if present, outperform both Batch- and LayerNorm as well as Conv layers. While the differences are small, they are statistically significant due to the large number of scores collected (see Fig. 16).

Second, we analyze whether the location of a layer inside a model plays a role, e.g., are earlier layers more interpretable than later ones? The average per-unit MIS (for each layer type) is shown in Fig. 6A as a function of the relative depth of the layer. A value of zero corresponds to the first and a value of one to the last layer analyzed. The scores are averaged in bins of equal count defined by the relative layer depth to enhance readability. The resulting curves all follow a similar, almost sinusoidal, pattern: They start high, decrease in the first fifth, then increase steadily until they drop in the last tenth again.

Third, it is interesting to probe the influence of the width of layers on their average interpretability. Based on the superposition hypothesis (Elhage et al., 2022; Olah et al., 2020; Arora et al., 2018; Goh, 2016), one might expect wider layers to be more interpretable as features do not have to form in superposition (i.e., as *polysemantic* units) but can arise in a disentangled form (i.e., as *monosemantic* units). Fig. 6B shows the relation between MIS and relative layer width. We use the relative rather than the absolute width to reduce the influence of the overall model and show the results of models with different architectures on the same axis. Note that, nevertheless, there might be other confounding factors correlated with the width, e.g., the layer depth. While we only see a moderate correlation for BatchNorm layers, we find a stronger one for Conv/Linear layers. It is unclear what causes this difference in behavior. However, we see this as a hint that one way to increase a model's interpretability is to increase the width (and not the number) of layers.

### 4.3 How Does the MIS Change During Training?

In the last set of experiments, we demonstrate how the MIS can be used to analyze models in a fine-grained way and obtain insights into their training dynamics. For this, we train a ResNet-50 on ImageNet-2012, following the training recipe A3 of Wightman et al. (2021), for 100 epochs.

Fig. 8 shows how the average per-unit MIS (left) changes during training. Notably, the initial MIS (of the untrained network) is already substantially above chance level. Nevertheless, during the first epoch, the MIS increases drastically to values around $0.93$. During the rest of the training, the score slowly decays. This indicates non-trivial dynamics of feature learning, which we analyze in Fig. 7. When showing the MIS as a function of top-1 accuracy during training (right), a strong anticorrelation (ignoring the first points) becomes evident. This aligns with the anticorrelation shown in Fig. 4A.

To better understand the dynamics through the training — most importantly during the first epoch — we zoom in to find out which units cause this strong change in MIS. Fig. 7 shows the change

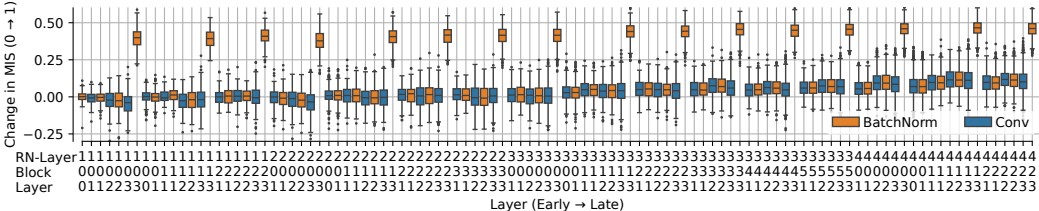

Fig. 7: **Change of Interpretability per Layer During Training.** To better understand the peak in interpretability after the first training epoch found in Fig. 8, we display the change in MIS during the first epoch, averaged over each layer. Note that layers are sorted by depth from left to right, and different colors encode different layer types. While the change in interpretability is moderately correlated with a layer's depth, we consistently see big improvements for the last BatchNorm layer of each block (i.e., *BatchNorm-\*-\*-3*). For a visualization covering the full training, see Fig. 18.

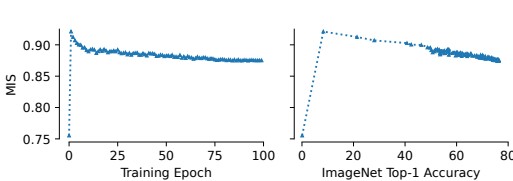

Figure 8: **Interpretability During Training.** For a ResNet-50 trained for 100 epochs on ImageNet, we track the MIS and accuracy after every epoch (epoch 0 refers to initialization). While the MIS improves drastically in the first epoch, it decays during the rest of the training (left). This results in an antiproportional relation between MIS and accuracy (right).

in MIS during the first epoch for each layer separately (ordered by their depth within the network). Surprisingly, we see that the change in MIS is dominated by a set of BatchNorm layers, namely the last ones of each ResNetBlock, whose MIS increases drastically. Moreover, we detect a small trend of later layers improving more strongly than earlier ones but generally do not see a difference between Conv and BatchNorm layers.

## 5 CONCLUSION

This paper presented the first fully automated interpretability metric for vision models — the machine interpretability score (MIS) — which is experimentally shown to be well aligned with human interpretability labels. We verified the alignment through both correlational and interventional experiments. We expect our MIS to enable experiments previously considered infeasible due to the costly reliance on human evaluations. To stress this, we demonstrated the metric's usefulness for formulating and testing new hypotheses about a network's behavior: Based on the largest comparison of vision models in terms of their per-unit interpretability so far, we investigated potential influences on their interpretability, such as a layer's depth and width. Most importantly, we find an anticorrelation between a model's downstream performance and its per-unit interpretability. Further, we performed the first detailed analysis of how the perceived interpretability changes during training.

While this paper considerably advances the state of interpretability evaluations, there are some open questions and potential future research directions. Most importantly, the performance of our MIS on a per-unit level is close to the noise ceiling determined by the limited number of human interpretability annotations available. This means that future changes in the MIS measure (e.g., based on other image perceptual similarities) might require additional human labels to determine the significance of performance improvements. Additional human labels could also be leveraged to improve the MIS by following Fu et al. (2023) to fine-tune the image similarity directly on human judgments. In another direction, using vision language models for computing the MIS could be interesting as this might also provide a textual description of a unit's sensitivity (Hernandez et al., 2022). Finding a differentiable approximation of the MIS will be valuable for explicitly training models to be interpretable (Zimmermann et al., 2023). Note that while this paper looked at the interpretability of channels and neurons, it can also be used for analyzing arbitrary directions in activation space. Thus, we expect the MIS to also be valuable for researchers generally looking for more interpretable representations of (artificial) neural activations (e.g., Graziani et al., 2023). Finally, exploring whether this concept of interpretability quantification can be expanded to LLMs is an exciting direction.

## AUTHOR CONTRIBUTIONS

RSZ led the project, which DK and RSZ initiated. DK proposed using perceptual similarity functions to build an interoperability metric. RSZ and WB conceived the final formulation of the metric. RSZ conducted all the experiments with suggestions from WB and feedback from DK. RSZ executed the data analysis, except for the estimation of the noise ceiling conducted by DK. RSZ created all the figures in the paper and wrote the manuscript with suggestions from DK and WB.

## ACKNOWLEDGMENTS

We thank Evgenia Rusak, Prasanna Mayilvahanan, Thaddäus Wiedemer, and Thomas Klein for their valuable feedback (in alphabetical order). This work was supported by the German Federal Ministry of Education and Research (BMBF): Tübingen AI Center, FKZ: 01IS18039A. WB acknowledges financial support via an Emmy Noether Grant funded by the German Research Foundation (DFG) under grant no. BR 6382/1-1 and via the Open Philantropy Foundation funded by the Good Ventures Foundation. WB is a member of the Machine Learning Cluster of Excellence, EXC number 2064/1 – Project number 390727645. This research utilized compute resources at the Tübingen Machine Learning Cloud, DFG FKZ INST 37/1057-1 FUGG. The authors thank the International Max Planck Research School for Intelligent Systems (IMPRS-IS) for supporting RSZ.

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

# A  DESCRIPTION OF THE 2-AFC TASK

## A.1  TASK DESIGN

Our proposed MIS builds on the 2-AFC task designed by Borowski et al. (2021) to conduct human psychophysics experiments. An example of such a task is given in Fig. 9.

This task aims to probe how well (human) participants can detect the sensitivity of a unit of a neural network based on visual explanations of it. Understanding the unit's sensitivity should allow participants to distinguish between a stimulus eliciting highly activating from one yielding low activation. Therefore, the task shows the participants two such images, called query images, and asks them to pick the image eliciting higher activation. To solve the task, participants also see two sets of visual explanations: Positive explanations describe the patterns the unit activates strongly for, while negative activations show patterns the unit weakly responds to. For solving this task, there are two potential strategies: Participants can either recognize a common pattern of the positive explanations in one of the query images, making this the correct choice. Or they detect a common pattern of the negative explanations in a query image, making the other one the right choice. See Borowski et al. (2021); Zimmermann et al. (2021) or Zimmermann et al. (2023) for alternative descriptions and visualizations of the task.

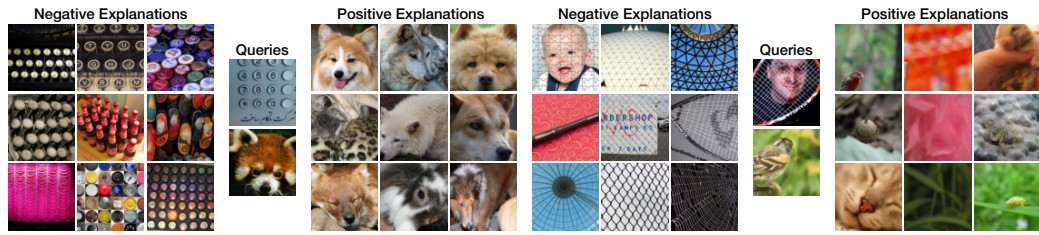

Fig. 9: **Examples of the 2-AFC Task.** For two different units of GoogLeNet one task each is shown. Every task contains a set of negative (left) and positive (right) visual explanations describing which visual feature the unit is sensitive to. In the center, two query images in the form of strongly and weakly activating dataset examples are shown, respectively. This means that each one of the two query images corresponds to the positive and the other to the negative explanations. The task is now to choose which query image corresponds to the positive ones.

## A.2  TASK CONSTRUCTION

For constructing tasks, we follow Zimmermann et al. (2023). Specifically, this means that we use $K = 9$ (positive and negative) explanations in each task. We restrict explanations to natural dataset examples to reduce complexity but note that the same setup can also be applied to other visual explanations, such as feature visualizations. To choose query images and explanations, we proceed as follows: For each unit, we determine the $N \cdot (K + 1)$ most and least activating images, respectively. Out of these, the $N \cdot K$ most extreme images are used as explanations, the others as query images. The $N \cdot K$ potential explanation images are uniformly distributed across tasks according to their elicited activation level (see (Borowski et al., 2021; Zimmermann et al., 2023) for more details).

# B    INFLUENCE OF THE UNDERLYING PERCEPTUAL SIMILARITY ON THE MACHINE INTERPRETABILITY SCORE

As stated in Sec. 3, we used DreamSim (Fu et al., 2023) as the underlying perceptual similarity $f$ for all experiments shown so far. We now repeat the experiments on IMI in Sec. 4.1.1 with two alternative similarity measures: LPIPS (Zhang et al., 2018) and DISTS (Ding et al., 2022). While all three measures are based on learned image features, DreamSim leverages an ensemble of modern vision models trained on larger datasets compared to LPIPS and DISTS, which use AlexNet (Krizhevsky et al., 2012) and VGG16 (Simonyan & Zisserman, 2015) trained on ImageNet, respectively. According to Fu et al. (2023), DreamSim clearly outperforms LPIPS and DISTS on image similarity benchmarks.

When comparing MIS based on DreamSim with one based on LPIPS and DISTS on a per-model level (see Fig. 10) one sees very similar results and strong correlations between each MIS and HIS. This might suggest that the choice of the similarity function to use has little influence on the quality of MIS. The picture, however, changes when zooming in and looking at per-unit interpretability (see Fig. 12). Now, it becomes evident that the MIS based on DreamSim outperforms that based on LPIPS and DISTS, indicated by the higher correlation and smaller spread of the point cloud. We, therefore, conclude that DreamSim is the best perceptual similarity available for computing machine interpretability scores.

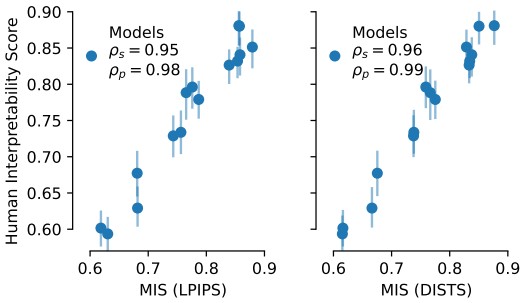

Fig. 10: **LPIPS and DISTS Perform Similarly as DreamSim when Comparing Models.** We compare DreamSim with two earlier perceptual similarity metrics, LPIPS and DISTS. All three lead to similar results on IMI (cf. Fig. 2A). See Fig. 12 for comparing these similarity functions on a per-unit level. standard deviation.

**Noise Ceiling of Annotations in IMI**    To put the difference in performance between the perceptual similarities on a per-unit level into context, we estimate the noise ceiling of the data: As the HIS for a single unit is a (potentially) noisy estimate over (up to 30) human decisions, it has some uncertainty. To take this into account, we run a statistical simulation, in which we model individual human responses as binary decisions from a Bernoulli distribution whose mean equals the unit's HIS. We can now simulate human decisions by sampling from the distribution. Then, we compute the correlation between MIS and simulated HIS and repeat the process 1 000 times. The resulting *noise ceiling* is compared to the correlations obtained when using LPIPS, DISTS, and DreamSim in Fig. 11. We see that DreamSim's performance is very close to the noise ceiling for estimating the per-unit human interpretability.

# C    SENSITIVITY OF THE MIS ON THE NUMBER OF TASKS

As described in Sec. 3, we compute the MIS by averaging over $N = 20$ tasks. This choice was initially motivated by previous work by Borowski et al. (2021). We investigate now how this choice influences the MIS. For this, we perform two experiments for GoogLeNet (see Fig. 13). First, we use the method for constructing tasks described before in Appx. A.2 to create 20 tasks per unit and then compute how the MIS changes when only using the first $i = 1, \ldots, 19$ tasks compared to all 20. While this setting is straightforward to analyze, it does not reflect how the number of tasks influences the MIS computation in practice: Using the task creation above, the chosen number of

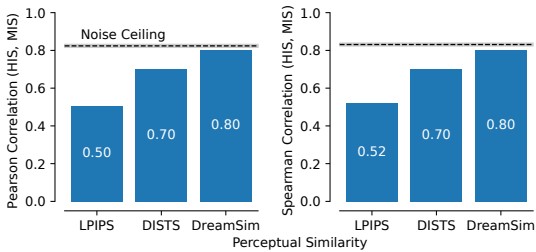

Fig. 11: **Best Perceptual Similarity Approaches Noise Ceiling.** Considering the noise ceiling, caused by the inherent uncertainty of the HIS, the best perceptual similarity (DreamSim) shows an almost perfect performance. The black bar and shaded area show the mean correlation and standard deviation over 1 000 simulations, respectively.

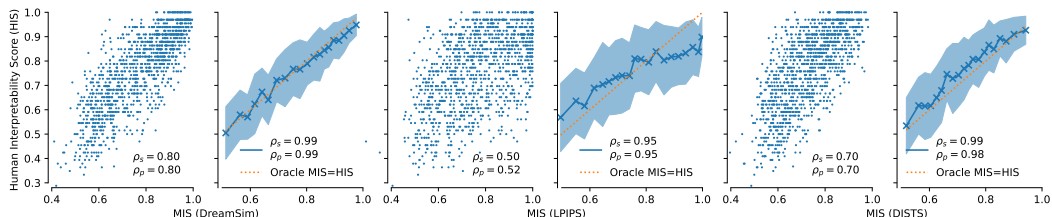

Fig. 12: **LPIPS and DISTS Perform Worse than DreamSim when Comparing Individual Units.** We compare DreamSim with two earlier perceptual similarity metrics, LPIPS and DISTS. While LPIPS and DISTS perform similarly to DreamSim on a per-model level of IMI (cf. Fig. 12), they lead to worse performance on a per-unit level.

tasks influences the creation of all tasks, e.g., adding one more task changes which images are used for previous tasks. Therefore, in the second experiment, we again measure how the MIS changes when using $i = 1, \ldots, 19$ tasks compared to 20, but recreate all tasks when increasing their number. For both settings, we see that the residual converges to zero, with a slower convergence in the more realistic setting.

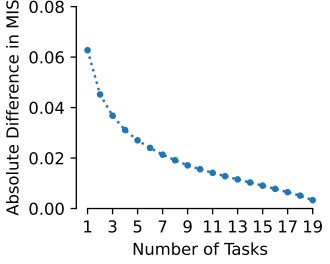 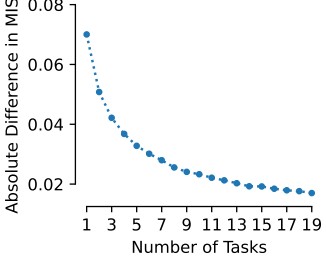

(a) New tasks do not influence earlier tasks.    (b) New tasks influence earlier tasks.

Fig. 13: **Convergence of MIS.** We investigate how MIS changes depending on the number of tasks $N$ that it is computed over. Here, we distinguish between two settings. In (a), we simulate that adding another task does not change the selection of query images and explanations in earlier tasks; in (b), this is not the case. While the former is easier to analyze due to a reduced level of randomness, note that the latter is the more relevant setting in practice. For both cases, we visualize the average absolute difference in MIS estimated for $< 20$ and $N = 20$ tasks.

## D    APPLYING MIS FOR DIFFERENT EXPLANATION METHODS

The experiments in Sec. 4 compute the MIS for one type of explanation, namely strongly activating dataset examples. We now demonstrate that the same approach easily generalizes to other visual explanations: feature visualizations. We do not tune any hyperparameters but re-use the same as presented in Sec. 3 for dataset examples as explanations. In Fig. 14 we repeat the experiment from Fig. 2A and again see a strong correlation between MIS and HIS.

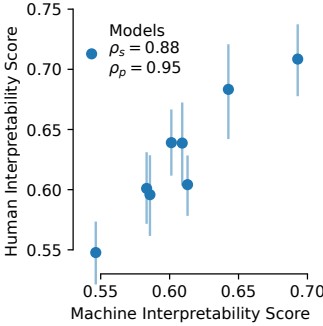

Fig. 14: **MIS Generalizes Well to Other Explanation Types.** We find a high correlation between MIS and HIS for other explanation types (feature visualizations). See Fig. 2A for the corresponding results for using natural dataset examples as explanations.

## E    ANALYSIS OF CONSTANT UNITS

After training a network, it might happen that some of its units effectively become non-active/constant for any relevant image. We here call a unit *constant* if the difference between maximally and minimally elicited activation by the entire ImageNet-2012 training set is less than $10^{-8}$. As mentioned at the beginning of Sec. 4, we excluded those units in our analysis, as they do not present any interesting behavior that is worth understanding. Note that this does not mean that it will not be interesting to understand why such units exist. In Fig. 15, we display the ratio of constant units for each model. For most models, we see a low number of constant units: Specifically, we see that out of the 835 models investigated, 256 do not contain any constant units, 89 contain more than 1 % and 22 more than 5 %. Note that we here used the same notion of units as in the rest of the paper, meaning that we take the spatial mean of feature maps with spatial dimensions (e.g., for convolutional layers).

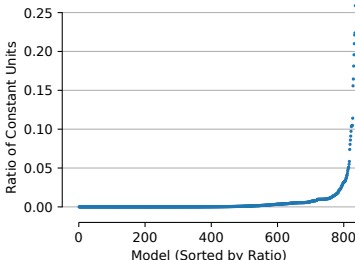

Fig. 15: **Ratio of Constant Units.** We compute the ratio of units constant with respect to the input (over the training set of ImageNet-2012) for all models considered. While the ratio is low for most models, it becomes large for a few models.

## F    DETAILS ON MODELS

In addition to the 9 models investigated by Zimmermann et al. (2023) (GoogLeNet, ResNet-50, Clip ResNet-50, Robust (L2) ResNet-50, DenseNet-101, WideResNet-50, Clip ViT-B32. ViT-B32), we include one more model suggested by them (Robust (L2) ResNet-50) and 825 models from timm (Wightman, 2019):

xcit_tiny_12_p16_224.fb_in1k, vit_tiny_patch16_384.augreg_in21k_ft_in1k, pit_xs_224.in1k, repghostnet_111.in1k, regnetz_c16_evos.ch_in1k, poolformer_m48.sail_in1k, repghostnet_080.in1k, volo_d3_448.sail_in1k, vit_base_patch16_224.augreg_in21k_ft_in1k, regnety_320.tv2_in1k, densenet121.ra_in1k, mobilenetv3_large_100.ra_in1k, repghostnet_150.in1k, seresnext26ts.ch_in1k, regnety_160.swag_ft_in1k, hr-net_w40.ms_in1k, convnext_small.in12k_ft_in1k, vit_base_patch16_224.sam_in1k, seresnextaa101d_32x8d.sw_in12k_ft_in1k_288, vit_tiny_r_s16_p8_384.augreg_in21k_ft_in1k, regnety_320.pycls_in1k, cs3darknet_m.c2ns_in1k, vit_tiny_patch16_224.augreg_in21k_ft_in1k, resnet101c.gluon_in1k, convnextv2_atto.fcmae_ft_in1k, flexivit_base.600ep_in1k, xcit_small_12_p16_384.fb_dist_in1k, mo-bilenetv2_050.lamb_in1k, flexivit_base.300ep_in1k, resnext50_32x4d.tv_in1k, resnet152.tv_in1k, seresnext26d_32x4d.bt_in1k, fbnetv3_g.ra2_in1k, poolformer_s36.sail_in1k, resnet101_32x8d.tv_in1k, rexnet_130.nav_in1k, efficientvit_b2.r224_in1k, con-vnext_small.fb_in22k_ft_in1k_384, resnet50_gn.a1h_in1k, eva02_small_patch14_336.mim_in22k_ft_in1k, regnety_032.ra_in1k, res2net50d.in1k, convit_small.fb_in1k, regnetx_160.pycls_in1k, convnextv2_large.fcmae_ft_in22k_in1k_384, tf_efficientnet_b0.ns_jft_in1k, pit_ti_224.in1k, volo_d1_384.sail_in1k, xcit_small_12_p8_384.fb_dist_in1k, dpn131.mx_in1k, resnext101_64x4d.gluon_in1k, densenet169.tv_in1k, resnet101d.ra2_in1k, repghostnet_200.in1k, resnet18.a2_in1k, xcit_small_12_p16_224.fb_in1k, pvt_v2_b3.in1k, dm_nfnet_f1.dm_in1k, vit_large_patch32_384.orig_in21k_ft_in1k, convnextv2_tiny.fcmae_ft_in22k_in1k_384, gcresnet50t.ra2_in1k, nf_regnet_b1.ra2_in1k, volo_d1_224.sail_in1k, resnet50.ram_in1k, hrnet_w18_small_v2.ms_in1k, convnext_base.clip_laion2b_augreg_ft_in1k, regnetx_160.tv2_in1k, sequencer2d_l.in1k, convnext_large.fb_in22k_ft_in1k, botnet26t_256.c1_in1k, gc_efficientnetv2_rw_t.agc_in1k, wide_resnet50_2.racm_in1k, halonet50ts.a1h_in1k, cspresnet50.ra_in1k, resnetv2_50d_evos.ah_in1k, tf_efficientnetv2_b3.in21k_ft_in1k, resnet152.gluon_in1k, lambda_resnet26rpt_256.c1_in1k, fastvit_sa24.apple_dist_in1k, xcit_medium_24_p8_384.fb_dist_in1k, repvit_m0_9.dist_450e_in1k, regnetx_320.pycls_in1k, seresnextaa101d_32x8d.sw_in12k_ft_in1k, efficientvit_b2.r288_in1k, con-vnext_tiny.in12k_ft_in1k, xcit_large_24_p16_384.fb_dist_in1k, resnetv2_50.a1h_in1k, coatnet_0_rw_224.sw_in1k, efficientnet_es_pruned.in1k, dla60_res2net.in1k, efficientformer_l7.snap_dist_in1k, cait_xxs24_224.fb_in1k, vit_small_patch16_224.augreg_in21k_ft_in1k, tf_efficientnet_cc_b1_8e.in1k, efficientb1_b1.r288_in1k, halonet26t.a1h_in1k, mixnet_m.ft_in1k, hrnet_w44.ms_in1k, regnety_160.tv2_in1k, xcit_nano_12_p8_384.fb_dist_in1k, seresnext101_32x8d.ah_in1k, efficientvit_b2.r256_in1k, vit_base_patch16_clip_224.laion2b_ft_in12k_in1k, tf_efficientnet_lite2.in1k, deit3_small_patch16_224.fb_in1k, hrnet_w18_ssld.paddle_in1k, tf_efficientnet_b2.aa_in1k, crossvit_15_dagger_240.in1k, deit3_small_patch16_224.fb_in22k_ft_in1k, haloregnetz_b.ra3_in1k, tf_efficientnetv2_b0.in1k, eca_nfnet_l0.ra2_in1k, twins_pcpvt_small.in1k, ecaresnet50t.ra2_in1k, fastvit_sa12.apple_dist_in1k, skresnext50_32x4d.ra_in1k, resnet50d.a2_in1k, vit_base_patch32_clip_224.laion2b_ft_in1k, resnetblur50.bt_in1k, vit_base_patch16_224.orig_in21k_ft_in1k, resnet50.a1h_in1k, hard-corenas_e.miil_green_in1k, coatnext_nano_rw_224.sw_in1k, resnet10t.c3_in1k, poolformerv2_m48.sail_in1k, tf_efficientnet_b1.aa_in1k, edgenext_base.usi_in1k, tf_efficientnet_es.in1k, tres-net_l.miil_in1k_448, resnet152.a1h_in1k, mixnet_s.ft_in1k, resnet50.am_in1k, rexnet_100.nav_in1k, xcit_large_24_p8_224.fb_dist_in1k, deit3_base_patch16_224.fb_in22k_ft_in1k, xcit_tiny_24_p8_384.fb_dist_in1k, coat_lite_medium_384.in1k, focalnet_small_srf.ms_in1k, vit_base_patch8_224.augreg_in21k_ft_in1k, convnext_tiny_hnf.a2h_in1k, visformer_small.in1k, vit_small_r26_s32_384.augreg_in21k_ft_in1k, vgg16_bn.tv_in1k, eca_nfnet_l1.ra2_in1k, xcit_small_12_p8_224.fb_in1k, beitv2_base_patch16_224.in1k_ft_in22k_in1k, cs3edgenet_x.c2_in1k, vit_base_patch16_clip_384.laion2b_ft_in12k_in1k, xcit_small_12_p16_224.fb_dist_in1k, convformer_b36.sail_in1k_384, bat_resnext26ts.ch_in1k, caformer_b36.sail_in1k, dla34.in1k, crossvit_18_dagger_240.in1k, tf_efficientnetv2_s.in21k_ft_in1k, focalnet_base_srf.ms_in1k, con-vformer_b36.sail_in22k_ft_in1k_384, resnet34.tv_in1k, resmlp_24_224.fb_distilled_in1k, convnext_base.clip_laion2b_augreg_ft_in12k_in1k, caformer_s18.sail_in1k_384, resnetaa50.a1h_in1k, beitv2_base_patch16_224.in1k_ft_in1k, convformer_m36.sail_in22k_ft_in1k, in-ception_resnet_v2.tf_ens_adv_in1k, mobilenetv2_110d.ra_in1k, resnext101_32x4d.fb_swsl_ig1b_ft_in1k, regnetx_008.tv2_in1k, con-vnext_small.in12k_ft_in1k_384, levit_conv_128.fb_dist_in1k, volo_d3_224.sail_in1k, nest_tiny_jx.goog_in1k, mobileone_s2.apple_in1k, fastvit_t8.apple_dist_in1k, halo2botnet50ts_256.a1h_in1k, mobilenetv2_140.ra_in1k, caformer_m36.sail_in1k, seresnet50.ra2_in1k, hard-corenas_d.miil_green_in1k, convformer_b36.sail_in1k, regnety_320.swag_ft_in1k, volo_d4_448.sail_in1k, tf_efficientnet_b2.ns_jft_in1k, sebotnet33ts_256.a1h_in1k, vit_small_patch32_224.augreg_in21k_ft_in1k, vit_base_patch224.sam_in1k, resnetv2_50d_gn.ah_in1k, mo-bileone_s4.apple_in1k, coat_small.in1k, tf_mixnet_l.in1k, resnet34.a2_in1k, regnetx_032.pycls_in1k, resnetaa101d.sw_in12k_ft_in1k, lcnet_100.ra2_in1k, repvgg_b1.rvgg_in1k, crossvit_15_240.in1k, edgenext_x_small.in1k, repvit_m1_5.dist_300e_in1k, hard-corenas_a.miil_green_in1k, efficientformer_l1.snap_dist_in1k, tf_mobilenetv3_large_075.in1k, hrnet_w18_small.ms_in1k, tf_efficientnet_b2.in1k, ghostnetv2_130.in1k, ecaresnet26t.ra2_in1k, fastvit_s12.apple_in1k, xcit_tiny_12_p8_224.fb_dist_in1k, tres-net_m.miil_in21k_ft_in1k, fastvit_sa24.apple_in1k, resnetrs200.tf_in1k, convnextv2_nano.fcmae_ft_in1k, resnet50.ra_in1k, resnet34.bt_in1k, regnety_002.pycls_in1k, focalnet_base_lrf.ms_in1k, dla102.in1k, regnetz_e8.ra3_in1k, pvt_v2_b0.in1k, xcit_medium_24_p8_224.fb_in1k, regnety_640.seer.ft_in1k, resnet200d.ra2_in1k, caformer_s36.sail_in1k_384, deit3_small_patch16_384.fb_in22k_ft_in1k, eca_resnext26ts.ch_in1k, vgg13.tv_in1k, tf_efficientnet_lite0.in1k, resnet50.b1k_in1k, dla60_res2next.in1k, repvit_m1_1.dist_300e_in1k, convnext_base.fb_in22k_ft_in1k, tf_efficientnet_cc_b0_4e.in1k, ese_vovnet19b_dw.ra_in1k, resnetv2_152x2_bit.goog_teacher_in21k_ft_in1k, deit_base_distilled_patch16_384.fb_in1k, resnet101d.gluon_in1k, convnext_large.fb_in22k_ft_in1k_384, darknet53.c2ns_in1k, poolformerv2_s36.sail_in1k, convformer_m36.sail_in22k_ft_in1k, gmlp_s16_224.ra3_in1k, convformer_s18.sail_in1k, effi-cientnet_em.ra2_in1k, inception_v3.gluon_in1k, resmlp_12_224.fb_in1k, tresnet_l.miil_in1k, ecaresnet101d_pruned.miil_in1k, resnet152.a2_in1k, vit_small_patch32_384.augreg_in21k_ft_in1k, inception_v3.tf_adv_in1k, repghostnet_130.in1k, levit_conv_384.fb_dist_in1k, repvit_m1_5.dist_450e_in1k, efficientnet_el.ra_in1k, seresnet50.a2_in1k, pit_s_distilled_224.in1k, cspdarknet53.ra_in1k, tf_efficientnet_cc_b0_8e.in1k, densenet201.tv_in1k, resnext50_32x4d.a1_in1k, cs3sedarknet_l.c2ns_in1k, cait_s24_384.fb_dist_in1k, spnas-net_100.rmsp_in1k, res2net50_14w_8s.in1k, repvgg_d2se.rvgg_in1k, regnetx_032.tv2_in1k, crossvit_18_dagger_408.in1k, pit_b_distilled_224.in1k, cs3darknet_focus_l.c2ns_in1k, resnet50.bt_in1k, vgg11.tv_in1k, convnextv2_femto.fcmae_ft_in1k, convnext_nano.in12k_ft_in1k, resnext101_64x4d.tv_in1k, convnext_nano.d1h_in1k, cspresnet50.ra_in1k, tf_mixnet_m.in1k, xcit_tiny_12_p16_384.fb_dist_in1k, seresnet50.a1_in1k, efficientnetv2_rw_t.ra2_in1k, regnety_032.tv_in1k, inception_resnet_v2.tf_in1k, eva_large_patch14_196.in22k_ft_in1k, pvt_v2_b1.in1k, convformer_m36.sail_in1k_384, densenet161.tv_in1k, dla102x.in1k, ed-genext_small_rw.sw_in1k, regnety_016.tv2_in1k, convnextv2_base.fcmae_ft_in1k, vit_large_patch14_clip_336.laion2b_ft_in12k_in1k, levit_conv_128s.fb_dist_in1k, hrnet_w48.ms_in1k, resnet101.a1h_in1k, xcit_medium_24_p8_224.fb_dist_in1k, resnetrs152.tf_in1k, convnextv2_nano.fcmae_ft_in22k_in1k, convnextv2_tiny.fcmae_ft_in22k_in1k, resnext50d_32x4d.bt_in1k, gernet_s.idstcv_in1k, selec-sls42b.in1k, repvit_m3.dist_in1k, resnest50d_1s4x24d.in1k, dpn98.mx_in1k, xcit_nano_12_p16_224.fb_in1k, regnetx_016.pycls_in1k, xcit_medium_24_p16_224.fb_in1k, caformer_s18.sail_in1k, sehalonet33ts.ra2_in1k, tinynet_c.in1k, xcit_tiny_24_p16_224.fb_dist_in1k, flexivit_small.300ep_in1k, resnext101_32x8d.tv2_in1k, convnextv2_base.fcmae_ft_in22k_in1k_384, semnasnet_075.rmsp_in1k, res2net50_26w_4s.in1k, cait_xxs24_384.fb_dist_in1k, mobilenetv2_120d.ra_in1k, seresnext26_32x4d.bt_in1k, flexivit_base.1200ep_in1k, res2net50_26w_6s.in1k, vit_base_patch16_clip_384.openai_ft_in1k, nest_base_jx.goog_in1k, ecaresnetlight.miil_in1k, repvgg_b0.rvgg_in1k, ecaresnet50t.a1_in1k, inception_next_tiny.sail_in1k, regnety_032.pycls_in1k, mixer_b16_224.miil_in21k_ft_in1k, poolformer_s12.sail_in1k, vit_base_patch32_clip_384.openai_ft_in12k_in1k, vit_base_patch32_384.augreg_in21k_ft_in1k, effi-cientvit_b1.r224_in1k, vit_base_patch16_clip_384.laion2b_ft_in1k, deit_small_distilled_patch16_224.fb_in1k, efficientvit_b0.r224_in1k, resnest50d.in1k, regnety_120.pycls_in1k, semnasnet_100.rmsp_in1k, wide_resnet50_2.tv_in1k, xcit_small_24_p16_224.fb_in1k, resnet101.a3_in1k, fastvit_t12.apple_in1k, tf_efficientnet_lite1.in1k, tinynet_a.in1k, resmlp_big_24_224.fb_distilled_in1k, cs3se_edgenet_x.c2ns_in1k, resnetv2_152x2_bit.goog_teacher_in21k_ft_in1k_384, resnext50_32x4d.tv2_in1k, efficient-net_b2.ra_in1k, convformer_s18.sail_in22k_ft_in1k_384, caformer_s18.sail_in22k_ft_in1k_384, deit3_base_patch16_224.fb_in1k, vit_base_patch32_clip_384.laion2b_ft_in12k_in1k, vit_medium_patch16_gap_384.sw_in12k_ft_in1k, sequencer2d_s.in1k, mo-bileone_s0.apple_in1k, edgenext_base.in21k_ft_in1k, deit3_medium_patch16_224.fb_in1k, efficientformerv2_l.snap_dist_in1k, lambda_resnet50ts.a1h_in1k, xception41p.ra3_in1k, resnext50_32x4d.a3_in1k, crossvit_small_240.in1k, repvgg_a1.rvgg_in1k, resnet51q.ra2_in1k, xcit_small_24_p16_384.fb_dist_in1k, vit_base_patch32_clip_224.openai_ft_in1k, flexivit_large.300ep_in1k, repvgg_b3g4.rvgg_in1k, resnext50_32x4d.a1h_in1k, coat_lite_medium.in1k, vit_base_patch32_clip_448.laion2b_ft_in12k_in1k,

resnext50_32x4d.gluon_in1k, repvgg_b2.rvgg_in1k, vit_base_patch16_rpn_224.sw_in1k, mixer_b16_224.goog_in21k_ft_in1k, resnet50.c2_in1k, lamhalobotnet50ts_256.a1h_in1k, tiny_vit_21m_512.dist_in22k_ft_in1k, xcit_large_24_p16_224.fb_dist_in1k, repvgg_a2.rvgg_in1k, gernet_l.idstcv_in1k, mobilevitv2_050.cvnets_in1k, convnextv2_base.fcmae_ft_in22k_in1k, resnet18.a3_in1k, ecaresnet50d.miil_in1k, coat_lite_small.in1k, convnext_xlarge.fb_in22k_ft_in1k, mobilevitv2_075.cvnets_in1k, cait_s36_384.fb_dist_in1k, efficientformerv2_s1.snap_dist_in1k, resnet18.fb_swsl_ig1b_ft_in1k, mobileone_s1.apple_in1k, resnet61q.ra2_in1k, tf_efficientnetv2_b3.in1k, mobilevitv2_175.cvnets_in1k, convnext_tiny.fb_in22k_ft_in1k_384, crossvit_tiny_240.in1k, caformer_b36.sail_in22k_ft_in1k_384, resnet152d.ra2_in1k, convit_base.fb_in1k, tinynet_b.in1k, deit3_large_patch16_384.fb_in22k_ft_in1k, regnetx_004_tv.tv2_in1k, cait_xxs36_384.fb_dist_in1k, convnext_nano_ols.d1h_in1k, efficientnet_lite0.ra_in1k, inception_v4.tf_in1k, hrnet_w18.ms_in1k, gernet_m.idstcv_in1k, convformer_s36.sail_in22k_ft_in1k_384, deit_tiny_distilled_patch16_224.fb_in1k, deit_small_patch16_224.fb_in1k, vit_large_patch14_clip_336.laion2b_ft_in1k, crossvit_18_240.in1k, resnet26.bt_in1k, deit3_base_patch16_384.fb_in22k_ft_in1k, convformer_s36.sail_in1k, convnext_small.fb_in22k_ft_in1k, selecsls60b.in1k, efficientnet_b0.ra_in1k, focalnet_tiny_srf.ms_in1k, ecaresnet101d.miil_in1k, regnetx_080.tv2_in1k, mobileone_s3.apple_in1k, mobilenetv3_rw.rmsp_in1k, poolformerv2_m36.sail_in1k, seresnextaa101d_32x8d.ah_in1k, levit_conv_192.fb_dist_in1k, focalnet_tiny_lrf.ms_in1k, regnety_320.swag_lc_in1k, tresnet_v2_l.miil_in21k_ft_in1k, seresnet50.a3_in1k, dla46x_c.in1k, cs3darknet_x.c2ns_in1k, tf_efficientnet_b0.ap_in1k, vit_base_patch16_224.augreg2_in21k_ft_in1k, resnext101_32x8d.fb_ssl_ig1b_1000m_ft_in1k, xcit_large_24_p8_384.fb_dist_in1k, tinynet_a.in1k, cait_xs24_384.fb_dist_in1k, fastvit_sa12.apple_in1k, hrnet_w64.ms_in1k, regnety_016.pycls_in1k, wide_resnet101_2.tv2_in1k, beitv2_large_patch16_224.in1k_ft_in22k_in1k, hrnet_w30.ms_in1k, resnet101.tv_in1k, repvit_m2.dist_in1k, coatnet_nano_rw_224.sw_in1k, flexivit_small.1200ep_in1k, tf_efficientnet_b0.in1k, tf_efficientnet_b1.in1k, efficientformer_l3.snap_dist_in1k, vit_base_patch16_384.augreg_in21k_ft_in1k, xcit_tiny_24_p8_224.fb_dist_in1k, dla102x2.in1k, hardcorenas_f.miil_green_in1k, regnety_064.ra3_in1k, resnext101_32x4d.gluon_in1k, tf_efficientnetv2_b2.in1k, resnet32ts.ra2_in1k, xcit_tiny_12_p8_384.fb_dist_in1k, inception_v3.tv_in1k, xcit_large_24_p16_224.fb_in1k, ecaresnet50t.a3_in1k, repvit_m2_3.dist_450e_in1k, fbnetv3_b.ra2_in1k, vit_base_patch8_224.augreg2_in21k_ft_in1k, cs3darknet_l.c2ns_in1k, convnext_base.clip_laion2b_augreg_ft_in12k_in1k_384, regnety_160.deit_in1k, regnety_160.pycls_in1k, dla60x.in1k, xcit_tiny_24_p16_384.fb_dist_in1k, eva02_tiny_patch14_336.mim_in22k_ft_in1k, volo_d2_224.sail_in1k, regnety_160.swag_lc_in1k, vit_base_patch32_clip_224.laion2b_ft_in12k_in1k, tf_mixnet_s.in1k, repvit_m1_0.dist_300e_in1k, convnextv2_large.fcmae_in1k, resmlp_12_224.fb_distilled_in1k, xcit_medium_24_p16_384.fb_dist_in1k, regnety_080_tv.tv2_in1k, dpn107.mx_in1k, inception_v3.tf_in1k, dpn68.mx_in1k, efficientnet_es.ra_in1k, mnasnet_100.rmsp_in1k, resnet101.tv2_in1k, res2next50.in1k, vit_base_patch16_clip_384.openai_ft_in1k, tf_efficientnet_b1.ns_jft_in1k, flexivit_small.600ep_in1k, visformer_tiny.in1k, resnet50.a1_in1k, dla60.in1k, regnetz_d32.ra3_in1k, senet154.gluon_in1k, efficientnetv2_rw_s.ra2_in1k, focalnet_small_lrf.ms_in1k, seresnet33ts.ra2_in1k, fbnetc_100.rmsp_in1k, resnet18d.ra2_in1k, resnet34.a3_in1k, dla60x_c.in1k, efficientnet_b1_pruned.in1k, efficientformerv2_s2.snap_dist_in1k, resnet50s.gluon_in1k, resnet101.a2_in1k, regnety_040.ra3_in1k, convmixer_1536_20.in1k, regnety_008_tv.tv2_in1k, resnet152.a1_in1k, mixnet_l.ft_in1k, gcresnet26ts.ch_in1k, vit_base_patch16_clip_224.openai_ft_in1k, fastvit_ma36.apple_in1k, vgg16.tv_in1k, gcresnet50ts.ch_in1k, xcit_tiny_12_p16_224.fb_dist_in1k, regnety_008.pycls_in1k, resmlp_36_224.fb_distilled_in1k, regnetz_040_h.ra3_in1k, inception_next_base.sail_in1k, dm_nfnet_f0.dm_in1k, resnet50.d_in1k, efficientnet_b2_pruned.in1k, resnet18.tv_in1k, rexnet_150.nav_in1k, convnext_large_mlp.clip_laion2b_soup_ft_in12k_in1k_320, ghostnetv2_160.in1k, vit_small_patch16_384.augreg_in21k_ft_in1k, convnext_xlarge.fb_in22k_ft_in1k_384, mobilenetv3_small_075.lamb_in1k, regnetz_d8_evos.ch_in1k, dm_nfnet_f3.dm_in1k, repvgg_b3.rvgg_in1k, convnext_large_mlp.clip_laion2b_augreg_ft_in1k_384, dpn68b.mx_in1k, resnext101_32x8d.fb_wsl_ig1b_ft_in1k, deit3_large_patch16_384.fb_in1k, convformer_s18.sail_in1k_384, repghostnet_058.in1k, fastvit_sa36.apple_dist_in1k, resnext50_32x4d.a2_in1k, regnetx_040.pycls_in1k, vit_base_r50_s16_384.orig_in21k_ft_in1k, vit_base_patch16_clip_224.laion2b_ft_in1k, deit3_base_patch16_384.fb_in1k, tf_efficientnetv2_s.in1k, ecaresnet50t.a2_in1k, resnetrs50.tf_in1k, gmixer_24_224.ra3_in1k, resnetaa50d.sw_in12k_ft_in1k, tresnet_xl.miil_in1k, resnest101e.in1k, regnetx_004.pycls_in1k, mnasnet_small.lamb_in1k, repvgg_a0.rvgg_in1k, resnetv2_50x1_bit.goog_in21k_ft_in1k, cait_s24_224.fb_dist_in1k, regnety_004.tv2_in1k, convnext_base.fb_in22k_ft_in1k_384, convnext_tiny.fb_in22k_ft_in1k, convnext_tiny.in12k_ft_in1k_384, eca_halonext26ts.c1_in1k, resnet18.gluon_in1k, fastvit_s12.apple_dist_in1k, deit_base_patch16_224.fb_in1k, hrnet_w18.ms_aug_in1k, resnet33ts.ra2_in1k, seresnext101_64x4d.gluon_in1k, convnext_small.fb_in1k, convformer_s36.sail_in1k_384, pit_ti_distilled_224.in1k, resnet50.tv2_in1k, nest_small_jx.goog_in1k, resmlp_36_224.fb_in1k, hrnet_w18_small.gluon_in1k, vit_base_patch16_384.augreg_in1k, resnet50.fb_swsl_ig1b_ft_in1k, poolformer_m36.sail_in1k, tf_mobilenetv3_small_100.in1k, regnety_040.pycls_in1k, gcresnet33ts.ra2_in1k, resnet101s.gluon_in1k, darknetaa53.c2ns_in1k, poolformerv2_s12.sail_in1k, resnext50_32x4d.fb_ssl_yfcc100m_ft_in1k, poolformerv2_s24.sail_in1k, eca_resnet33ts.ra2_in1k, repvit_m2_3.dist_300e_in1k, nf_resnet50.ra2_in1k, convnext_pico_ols.d1_in1k, caformer_s36.sail_in1k, regnetz_040.ra3_in1k, vit_small_r26_s32_224.augreg_in1k_ft_in1k, resnext26ts.ra2_in1k, mixnet_xl.ra_in1k, deit_base_patch16_384.fb_in1k, repvit_m1_0.dist_450e_in1k, convmixer_1024_20_ks9_p14.in1k, regnety_064.pycls_in1k, resnet34.gluon_in1k, res2net101_26w_4s.in1k, nfnet_l0.ra2_in1k, resnet34d.ra2_in1k, convnextv2_nano.fcmae_ft_in22k_in1k_384, twins_pcpvt_base.in1k, resnetv2_101.a1h_in1k, xcit_nano_12_p8_224.fb_dist_in1k, xcit_small_24_p8_224.fb_dist_in1k, resnet50.b2k_in1k, deit3_small_patch16_384.fb_in1k, hardcorenas_c.miil_green_in1k, coat_lite_mini.in1k, resnet152.tv2_in1k, densenetblur121d.ra_in1k, hrnet_w18_small_v2.gluon_in1k, vit_base_patch16_384.orig_in21k_ft_in1k, xcit_small_12_p8_224.fb_dist_in1k, convformer_m36.sail_in1k, xcit_nano_12_p16_384.fb_dist_in1k, resnet34.a1_in1k, convnext_atto_ols.a2_in1k, resnet14t.c3_in1k, twins_pcpvt_large.in1k, resnest26d.gluon_in1k, mobilenetv3_small_100.lamb_in1k, efficientnet_b3_pruned.in1k, vit_small_patch16_224.augreg_in1k, convnext_tiny.fb_in1k, resnet50d.a3_in1k, mobilevitv2_175.cvnets_in22k_ft_in1k, deit3_medium_patch16_224.fb_in22k_ft_in1k, seresnext101_32x4d.gluon_in1k, hardcorenas_b.miil_green_in1k, caformer_m36.sail_in22k_ft_in1k, ghostnetv2_100.in1k, ecaresnet50d_pruned.miil_in1k, caformer_s36.sail_in22k_ft_in1k_384, deit_tiny_patch16_224.fb_in1k, fastvit_sa36.apple_in1k, regnety_320.seer_ft_in1k, edgenext_small.usi_in1k, resmlp_big_24_224.fb_in22k_ft_in1k, regnety_160.lion_in12k_ft_in1k, regnety_160.sw_in12k_ft_in1k, tf_efficientnet_b1.ap_in1k, res2net50_48w_2s.in1k, eca_botnext26ts_256.c1_in1k, xcit_small_24_p8_224.fb_in1k, crossvit_9_dagger_240.in1k, coat_lite_tiny.in1k, resnetv2_101x1_bit.goog_in21k_ft_in1k, convnext_large_mlp.clip_laion2b_augreg_ft_in1k, xcit_nano_12_p16_224.fb_dist_in1k, cs3darknet_focus_m.c2ns_in1k, wide_resnet50_2.tv2_in1k, vit_base_patch16_clip_224.openai_ft_in12k_in1k, skresnet34.ra_in1k, repvgg_b1g4.rvgg_in1k, vgg19_bn.tv_in1k, repghostnet_100.in1k, regnetv_064.ra3_in1k, mobilenetv2_100.ra_in1k, convnext_femto.d1_in1k, resnet26t.ra2_in1k, regnetv_040.ra3_in1k, skresnet18.ra_in1k, caformer_m36.sail_in22k_ft_in1k_384, vit_base_patch32_384.augreg_in1k, regnetz_b16.ra3_in1k, hrnet_w48_ssld.paddle_in1k, resnest50d_4s2x40d.in1k, cait_xxs36_224.fb_dist_in1k, regnetx_016.tv2_in1k, xcit_small_24_p8_384.fb_dist_in1k, vit_tiny_r_s16_p8_224.augreg_in21k_ft_in1k, coat_mini.in1k, xcit_small_24_p16_224.fb_dist_in1k, caformer_s36.sail_in22k_ft_in1k, poolformer_s24.sail_in1k, resmlp_big_24_224.fb_in1k, regnetx_120.pycls_in1k, regnetz_d8.ra3_in1k, resnet50d.ra2_in1k, repvit_m1.dist_in1k, eca_nfnet_l2.ra3_in1k, resnet50d.gluon_in1k, seresnext50_32x4d.racm_in1k, vit_small_patch16_384.augreg_in1k, coat_tiny.in1k, xcit_nano_12_p8_224.fb_in1k, crossvit_base_240.in1k, resnet50d.a1_in1k, convformer_s36.sail_in22k_ft_in1k, convnextv2_large.fcmae_ft_in22k_in1k, resnet50.tv_in1k, resnet50.c1_in1k, pit_xs_distilled_224.in1k, efficientnet_b1.ft_in1k, tf_efficientnet_el.in1k, hrnet_w32.ms_in1k, vit_base_patch16_224.miil_in21k_ft_in1k, cs3sedarknet_x.c2ns_in1k, dpn68b.ra_in1k, tf_efficientnetv2_b1.in1k, regnety_004.pycls_in1k, tf_mobilenetv3_large_minimal_100.in1k, resnetrs101.tf_in1k, ese_vovnet39b.ra_in1k, mixer_l16_224.goog_in21k_ft_in1k, repghostnet_050.in1k, repvgg_b2g4.rvgg_in1k, repvit_m1_1.dist_450e_in1k, vit_base_patch32_224.augreg_in21k_ft_in1k, tf_mobilenetv3_large_100.in1k, pit_s_224.in1k, caformer_s18.sail_in22k_ft_in1k, wide_resnet101_2.tv_in1k, fastvit_t12.apple_dist_in1k, convmixer_768_32.in1k, vit_base_patch32_224.augreg_in1k, efficientformerv2_s0.snap_dist_in1k, resnest200e.in1k, levit_conv_256.fb_dist_in1k, resnet18.fb_ssl_yfcc100m_ft_in1k, vgg13_bn.tv_in1k, resnet152c.gluon_in1k, dla169.in1k, crossvit_15_dagger_408.in1k, convnext_femto_ols.d1_in1k, convnext_large.fb_in1k, regnetx_064.pycls_in1k, fastvit_t8.apple_in1k, seresnet152d.ra2_in1k, vgg19.tv_in1k, vgg11_bn.tv_in1k, dm_nfnet_f2.dm_in1k, seresnext101d_32x8d.ah_in1k, inception_next_base.sail_in1k_384, lambda_resnet26t.c1_in1k, resnetv2_152x2_bit.goog_in21k_ft_in1k, fastvit_ma36.apple_dist_in1k, regnety_006.pycls_in1k, regnety_080.pycls_in1k, resnet50.fb_ssl_yfcc100m_ft_in1k, tf_mobilenetv3_small_075.in1k, regnetz_c16.ra3_in1k, edgenext_xx_small.in1k, crossvit_9_240.in1k, xcit_tiny_24_p8_224.fb_in1k, regnety_080.ra3_in1k, effi-

cientvit_b1.r256_in1k, tinynet_d.in1k, caformer_b36.sail_in1k_384, pvt_v2_b2.in1k, resnet26d.bt_in1k, convnext_pico.d1_in1k, pit_b_224.in1k, convnextv2_pico.fcmae_ft_in1k, fbnetv3_d.ra2_in1k, flexivit_large.1200ep_in1k, resnet50c.gluon_in1k, regnetx_080.pycls_in1k, convnext_base.fb_in1k, tf_efficientnet_em.in1k, vit_base_patch16_224.augreg_in1k, convit_tiny.fb_in1k, resnext50_32x4d.fb_swsl_ig1b_ft_in1k, dm_nfnet_f4.dm_in1k, resnet50.a3_in1k, convnext_atto.d2_in1k, efficientnet_el_pruned.in1k, volo_d2_384.sail_in1k, resnext101_32x4d.fb_ssl_yfcc100m_ft_in1k, repvit_m0_9.dist_300e_in1k, regnety_120.sw_in12k_ft_in1k, beit_base_patch16_384.in22k_ft_in22k_in1k, mobilenetv3_large_100.miil_in21k_ft_in1k, tf_efficientnet_b0.aa_in1k, inception_next_small.sail_in1k, deit_base_distilled_patch16_224.fb_in1k, lcnet_075.ra2_in1k, xcit_tiny_12_p8_224.fb_in1k, resnet101.gluon_in1k, dpn92.mx_in1k, resnet101.a1_in1k, selecsls60.in1k, beit_base_patch16_224.in22k_ft_in22k_in1k, convnextv2_tiny.fcmae_ft_in1k, res2net50_26w_8s.in1k, sequencer2d_m.in1k, vit_medium_patch16_gap_256.sw_in12k_ft_in1k, regnetx_008.pycls_in1k, resnet50.a2_in1k, res2net101d.in1k, vit_large_patch16_384.augreg_in21k_ft_in1k, pvt_v2_b2_li.in1k, regnetx_006.pycls_in1k, xcit_tiny_24_p16_224.fb_in1k, pvt_v2_b5.in1k, resnext50_32x4d.ra_in1k, resnest14d.gluon_in1k, caformer_m36.sail_in1k_384, resnet50.gluon_in1k, resnet152s.gluon_in1k, flexivit_large.600ep_in1k, resnetv2_50x1_bit.goog_distilled_in1k, resmlp_24_224.fb_in1k, deit3_large_patch16_224.fb_in1k, seresnext50_32x4d.gluon_in1k, densenet121.tv_in1k, resnet152.a3_in1k, ghostnet_100.in1k, tf_efficientnet_b2.ap_in1k, regnetx_002.pycls_in1k.

# G ADDITIONAL RESULTS

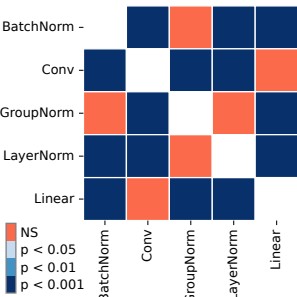

Fig. 16: **Differences Between Layer Types are Significant.** We analyze and test for statistical significances in the differences in MIS between different layer types (see Fig. 5. The reported significance levels were computed using Conover's test over the per-model and per-layer-type means with Holm's correction for multiple comparisons.

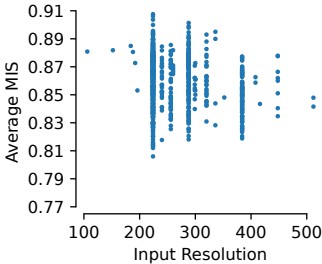

Fig. 17: **Influence of Input Resolution of MIS.** We show the average MIS per model as a function of the model's input resolution. No trend is apparent; models with the same resolution yield different interpretability levels.

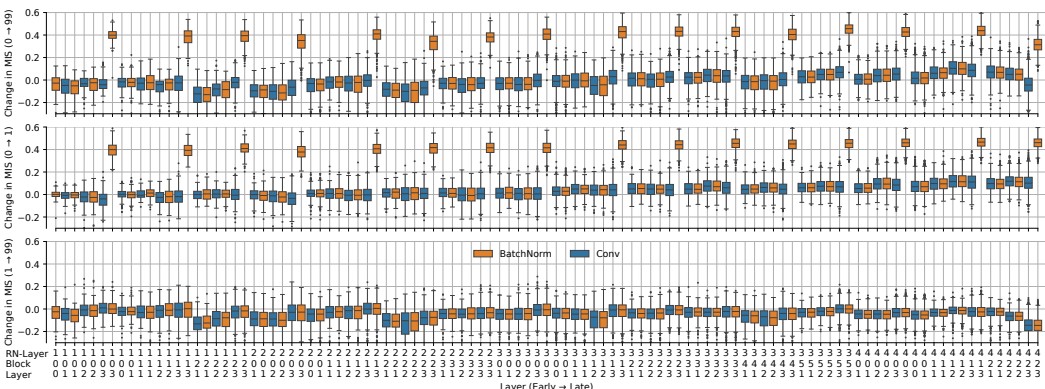

Fig. 18: **Change of Interpretability per Layer During Training.** Detailed version of Fig. 7.

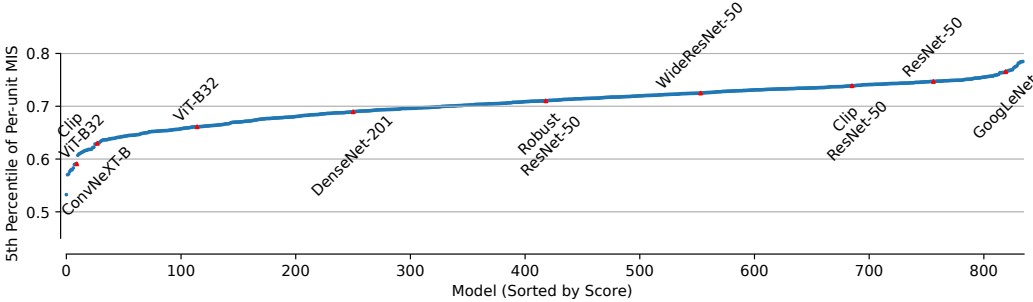

Fig. 19: **Comparison of the Minimum of the Per-unit MIS for Models.** While the mean of the per-unit interpretability varies in a rather narrow value range (see Fig. 3), we investigate differences in the distribution of scores. Specifically, we are interested in the effective width of the distribution, i.e., how low does the minimal MIS per model go? To make the analysis robust against outliers, we do not use the minimum but instead the 5th percentile. Note that this corresponds to the lower end of the shaded area in Fig. 3. Compared to the average MIS, we see higher variability across models.

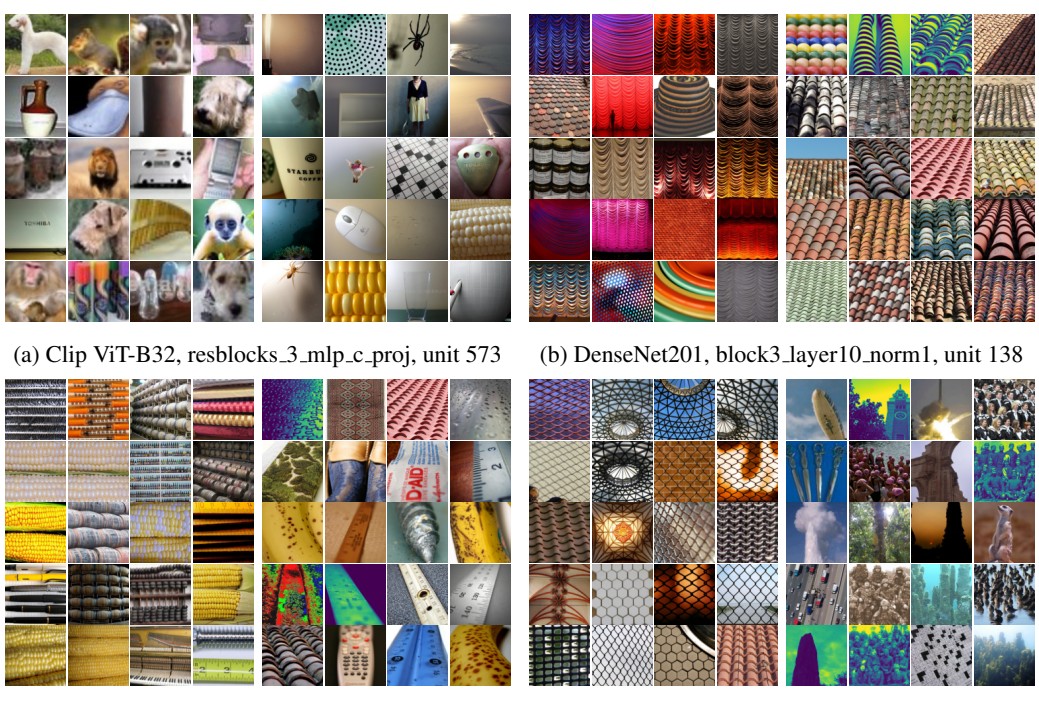

(a) Clip ViT-B32, resblocks_3_mlp_c_proj, unit 573    (b) DenseNet201, block3_layer10_norm1, unit 138

(c) DenseNet201, block3_layer29_conv1, unit 39    (d) DenseNet201, block3_layer35_norm2, unit 123

Fig. 20: **Visualization of Units for which MIS overestimates HIS.** To showcase the shortcomings of the MIS, we visualize four units for which the MIS predicts an interpretability that is higher than the measured HIS in Fig. 2B. See Fig. 21 for the opposite direction. For each unit, we show the 20 most (right) and 20 least (left) activating dataset exemplars.

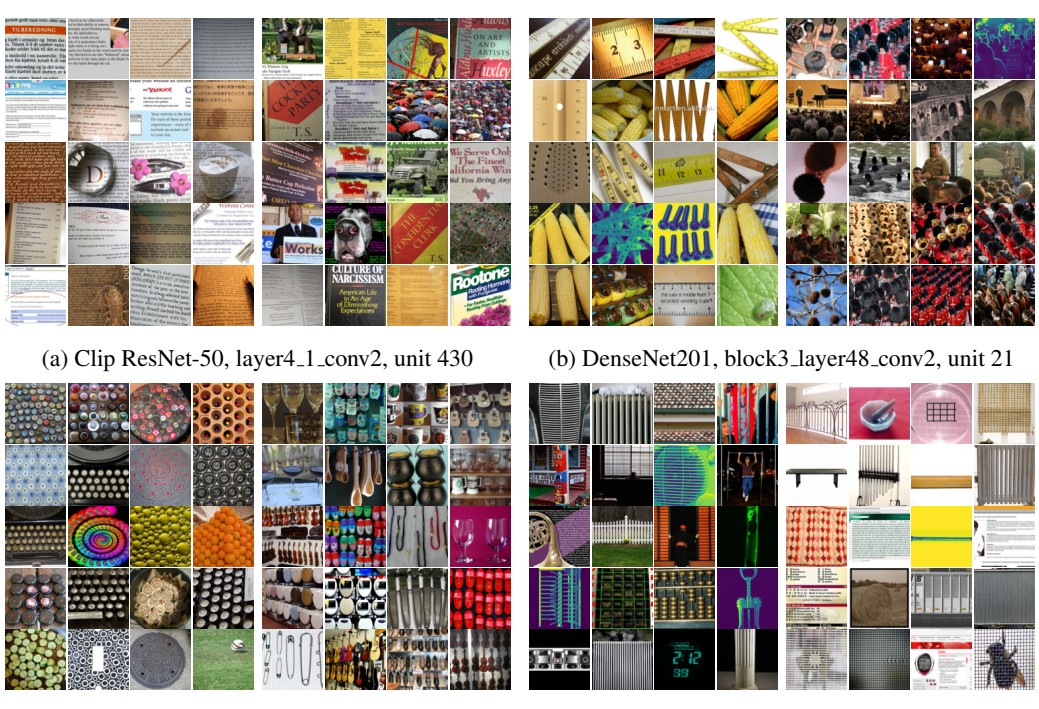

(a) Clip ResNet-50, layer4_1_conv2, unit 430     (b) DenseNet201, block3_layer48_conv2, unit 21

(c) DenseNet201, block3_layer48_norm1, unit 1369     (d) ViT-B32, block0_norm2, unit 358

Fig. 21: **Visualization of Units for which MIS underestimates HIS.** To showcase the shortcomings of the MIS, we visualize four units for which the MIS predicts an interpretability that is lower than the measured HIS in Fig. 2B. See Fig. 20 for the opposite direction. For each unit, we show the 20 most (right) and 20 least (left) activating dataset exemplars.

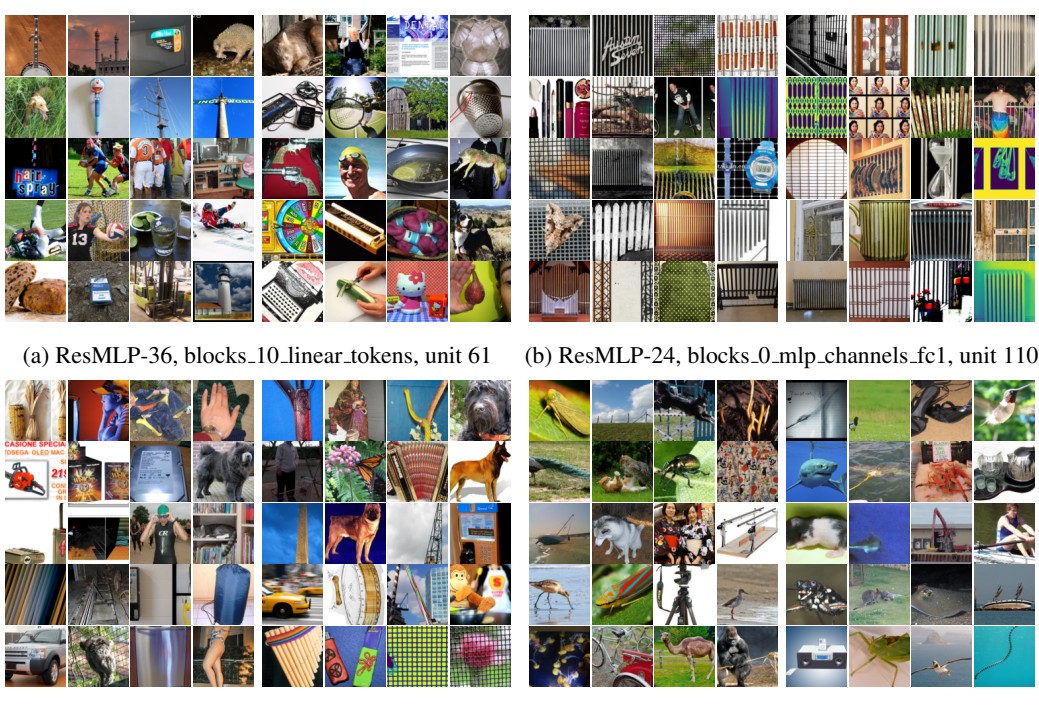

(a) ResMLP-36, blocks_10_linear_tokens, unit 61     (b) ResMLP-24, blocks_0_mlp_channels_fc1, unit 110

(c) GMixer-24, blocks_5_mlp_tokens_fc1, unit 166     (d) ResMLP-12, blocks_7_linear_tokens, unit 127

Fig. 22: **Visualization of Hard Units from Models with High Variability.** For the four models with the highest variability in MIS (see Fig. 4B), we visualize one of the units with the lowest MIS each. For each unit, we show the 20 most (right) and 20 least (left) activating dataset exemplars.

