# OpenReview forum: "Measuring Mechanistic Interpretability at Scale Without Humans"
_ICLR.cc/2024/Workshop/Re-Align — ICLR 2024 Workshop Re-Align Poster_

### Official Review · Reviewer_qp3U · 2024-02-20
**An interesting paper on an automated metric for the interpretability of vision models offers valuable insights into model training and architecture design.**

**Rating:** 3
**Fit:** 3
**Confidence:** 2

**Workshop Review:**

The paper presents the machine interpretability score (MIS), an automated metric for the interpretability of vision models.
The heart of the method is a predicted probability metric expressing how close an explanation is to a query. The visual explanations are a set of positive and negative images w.r.t. some unit; the queries are images with the most extreme positive and negative activation.
The paper provides a related work section on 1) “units” - the aspect of the NN methods striving to interpret, 2) quantifying interpretability, namely 2-AFC, and 3) automated interpretability research.
The author presents their method and argues for the hyperparameter choices, followed by a validation of the MIS.
The author compares MIS with the Human Interpretability Score (HIS), showing a strong correlation on the IMI dataset.
The high performance of MIS is further evaluated on different models, for different layers, and during training.
They make some interesting findings possible through the automated nature of MIS, e.g., that linear layers score higher than others (like conv layers).
The paper concludes with a reflection on the metric usefulness, a list of open questions, and future research directions.

Pros:
- The paper is well-written, and design and experiment decisions are easily understandable
- MIS can offer valuable insights that could benefit various research directions, as presented in the paper. I think I am most excited to use this metric to guide neural architecture choice.

Cons:
- One downside, which may be resolved after thinking more about this method, is its failure cases. For example, one thing that immediately struck me as suspicious is the high MIS for untrained models in Figure 12. I would expect a ~random chance for a model explanation to be correct. (Again, I may be lacking in grasping some crucial aspects of MIS.) I am happy to see Fig. 24/25 starting this discussion.

In summary, this is a fascinating paper, and I am happy to see more research like it, especially for language models!

**Reason For Not Giving Higher Score:**

N/A

**Reason For Not Giving Lower Score:**

- good idea, well executed

**Reviewer Domain:**

machine learning

---

### Official Review · Reviewer_1KN2 · 2024-02-24
**Review of submission 16**

**Rating:** 3
**Fit:** 3
**Confidence:** 3

**Workshop Review:**

This paper introduces the machine interpretability score (MIS), a metric that can be used to assess the interpretability of units in neural network layers. The quality of the paper is high and the results are particularly interesting for this workshop. Among the major contributions, the authors introduce a metric to automate the extraction of similarity explanations without requiring human annotations. Some results are highlighted: 1) MIS is strongly correlated with the human interpretability score (HIS), but it does not require annotation. 2) The MIS score is evaluated on more than 800 neural models on all layers, and these neural networks trained on Imagenet show an interesting anticorrelation between task accuracy and MIS. 3) The layer type, width, and depth in the NN are relevant for increasing per-unit interpretability. All these results are very interesting and merit to be discussed further at the workshop.

I have some remarks for the authors, which could be useful to address in future work or for a conference paper. The following does not count in my evaluation. I noticed that the related work focuses more on mechanistic interpretability rather than explainability, but this lacks a huge piece of research that is going on. The type of explanations that are used are prototype-like explanations (based on a similarity score) that are well known in the literature, see e.g. Guidotti et al. (2018) A survey of methods for explaining black box models. Also, I am not pretty much convinced that explanations of this kind (divided into positive and negative samples) are exhaustive, they only capture binary relations.
It would be interesting to explore other cases and also connect to causal alignment, see Geiger et al. (2021), Causal abstractions of neural networks.

**Reason For Not Giving Higher Score:**

N/A

**Reason For Not Giving Lower Score:**

N/A

**Reviewer Domain:**

machine learning

---

### Official Review · Reviewer_CHAJ · 2024-02-24
**Great paper**

**Rating:** 3
**Fit:** 3
**Confidence:** 2

**Workshop Review:**

First, I would like to congratulate the authors. The paper is really well-written, clear, and offers many interesting insights. I truly enjoyed reading it and believe it could significantly impact the explainability community. I particularly appreciate the interventional study.

I didn’t catch any major concerns (**M**), however, I've identified some minor concerns (**m**) and suggestions that could further enhance the paper's value.

**m.1**: Perhaps the most significant minor concern I have is that the paper relies exclusively on neuron-wise study. The authors acknowledge that superposition may strongly affect their data (and even measure it with their width experiments). Thus, I believe there is a subtle contradiction in your conclusion of fig.7: could it be that more performant models have more concepts and thus exhibit more superposition? I would really love to see a control, as I believe we may have a confounding factor: better-performing models may just be less monosemantic, not less interpretable. In other words, better models learn abstractions that push them to group images that are not perceptually similar. Sorry if I am being picky, but I believe this paper could have a strong impact, and this clarification could strengthen the argument or provide a clearer focus for the study's objectives.

**m.2**: I would be really interested in seeing examples where MIS fails to predict human scores. For example, are human and MIS able to "understand" low-freq/high-freq detectors like those in Schubert & al [1]? It's crucial to consider if the goal of explainable AI (XAI) is solely to uncover concepts/things humans already know, or if the most intriguing cases arise when images surprise us or differ significantly from expectations.
Put another way, someone could even argue that the neurons grouping images that look perceptually dissimilar may be the most interesting ones? I believe this is a minor issue, as we could use MIS to directly target those neurons, but I would love to see a discussion on that.

**m.3**: Figure 10 suggests that wider layers are more interpretable with less superposition, aligning with the hypothesis. However, it raises a question: is the paper merely measuring superposition? The findings suggest that as models become more performant and data becomes more compressed, superposition occurs, and neuron-wise interpretability diminishes because neurons may not be the correct basis for interpretation (see M.1).
The paper might benefit from explicitly testing if monosemanticity is what the benchmark measures. One approach could be training a model with an L1 penalty on the last neuron to bias the model towards a neuron basis to see if it achieves better performance on the benchmark. This could provide a direct test of the paper's underlying assumptions and contribute to the ongoing discussion about the role of monosemanticity in model interpretability.

**m.5**: Some missing citations for "Quantifying Interpretability" [2,3,4,5,6,7,8,9], as well as for the "interpretable representations of neural activations" [10,11,12,13,14,15]

Again, the issues raised are intended to offer constructive criticism and actionable feedback, reflecting my belief in the paper's potential to have a big positive impact on the community.


References:
- [1] Schubert, et al., "High-Low Frequency Detectors", Distill, 2021.
- [2] Nguyen et al., "The effectiveness of feature attribution methods and its correlation with automatic evaluation scores", NeurIPS.
- [3] Colin et al., "What I Cannot Predict, I Do Not Understand: A Human-Centered Evaluation Framework for Explainability Methods", NeurIPS.
- [4] Hase et al., "Evaluating Explainable AI: Which Algorithmic Explanations Help Users Predict Model Behavior?", ACL.
- [5] Nguyen et al., "Visual correspondence-based explanations improve AI robustness and human-AI team accuracy", NeurIPS.
- [6] Aodha et al., "Teaching categories to human learners with visual explanations", CVPR.
- [7] Chandrasekaran et al., "Do explanations make VQA models more predictable to a human?", ACL.
- [8] Alufaisan et al., "Does Explainable Artificial Intelligence Improve Human Decision Making?", AAAI.
- [9] Felix Biessmann et al., "Quality Metrics for Transparent Machine Learning With and Without Humans In the Loop Are Not Correlated", ICML.
- [10] Ghorbani et al., "Towards automatic concept-based explanations", NeurIPS.
- [11] Zhang et al., "Invertible concept-based explanations for cnn models with non-negative concept activation vectors", AAAI.
- [12] Fel et al., "Craft: Concept recursive activation factorization for explainability", CVPR.
- [13] Zhang et al., "Efficient neural network robustness certification with general activation functions", NeurIPS.
- [14] Vielhaben et al., "Multi-dimensional concept discovery (mcd): A unifying framework with completeness guarantees", TMLR.
- [15] Fel et al., "A Holistic Approach to Unifying Automatic Concept Extraction and Concept Importance Estimation", NeurIPS.

**Reason For Not Giving Higher Score:**

HIghest possible. I would like to recommend a talk for this paper.

**Reason For Not Giving Lower Score:**

No major concern

**Reviewer Domain:**

machine learning

---

### Decision · Program_Chairs · 2024-03-02

Accept (Poster)